# Critical Offset Magnetic PArticle Spectro-Scopy for rapid and highly sensitive medical point-of-care diagnostics

Patrick Vogel [1] ✉, Martin Andreas Rückert[1], Bernhard Friedrich[2], Rainer Tietze[2], Stefan Lyer[3], Thomas Kampf[1,4], Thomas Hennig[5], Lars Dölken [5,6], Christoph Alexiou[2] & Volker Christian Behr [1]

Magnetic nanoparticles (MNPs) have been adapted for many applications, e.g., bioassays for the detection of biomarkers such as antibodies, by controlled engineering of specific surface properties. Specific measurement of such binding states is of high interest but currently limited to highly sensitive techniques such as ELISA or flow cytometry, which are relatively inflexible, difficult to handle, expensive and time-consuming. Here we report a method named COMPASS (**C**ritical-**O**ffset-**M**agnetic-**Pa**rticle-**S**pectro**S**copy), which is based on a critical offset magnetic field, enabling sensitive detection to minimal changes in mobility of MNP ensembles, e.g., resulting from SARS-CoV-2 antibodies binding to the S antigen on the surface of functionalized MNPs. With a sensitivity of 0.33 fmole/50 μl (≙7 pM) for SARS-CoV-2-S1 antibodies, measured with a low-cost portable COMPASS device, the proposed technique is competitive with respect to sensitivity while providing flexibility, robustness, and a measurement time of seconds per sample. In addition, initial results with blood serum demonstrate high specificity.

The characterization of ensembles of magnetic nanoparticles (MNP) is a dynamically developing field that has found applications in many fields of research such as medicine, cancer theranostics, biosensing, catalysis, agriculture, and the environment[1–3]. Thus, a huge portfolio of different methods and techniques is available today to investigate the complex dynamics of MNP ensembles[4,5].

Magnetic particle spectroscopy (MPS) is a quite young technology for the characterization of MNPs. It uses an oscillating magnetic field of sufficient field strength to drive the MNP ensemble periodically into their non-linear magnetization response[6]. This reveals specific information for each MNP type in the form of higher harmonics of the

excitation frequency and can be used to measure parameters such as hydrodynamic diameter or viscosity, temperature of the surrounding solution, as well as the conjugations of chemical or biological compounds on the surface of the MNPs. In short, MPS is able to investigate the mobility of MNPs[7].

The fact that MPS directly measures the analytical signals from the entire sample volume makes bioassays simple and fast[8–10]. For example, functionalization of the surface of the MNPs by anchoring linkers, such as specific antibodies, allows the detection of viral proteins by binding specific epitopes. Cross-linking between the MNPs influences their mobility resulting in a minimal signal change. This enables the

[1]Department of Experimental Physics 5 (Biophysics), Julius-Maximilians-University Würzburg, Würzburg, Germany. [2]Department of Otorhinolaryngology, Head and Neck Surgery, Section of Experimental Oncology and Nanomedicine (SEON), Else Kröner-Fresenius-Stiftung-Professorship, University Hospital Erlangen, Erlangen, Germany. [3]Department of Otorhinolaryngology, Head and Neck Surgery, Section of Experimental Oncology and Nanomedicine (SEON), Professorship for AI-Controlled Nanomaterials, University Hospital Erlangen, Erlangen, Germany. [4]Department of Diagnostic and Interventional Neuroradiology, University Hospital Würzburg, Würzburg, Germany. [5]Institute for Virology and Immunobiology, Julius-Maximilians-University Würzburg, Würzburg, Germany. [6]Helmholtz Institute for RNA-based Infection Research, Helmholtz-Center for Infection Research, Würzburg, Germany. ✉ e-mail: Patrick.Vogel@physik.uni-wuerzburg.de

detection of, e.g., 44 nM H1N1 nucleoprotein or 1.56 nM SARS-CoV-2 spike protein within a measurement time of about 10 s[8,10]. However, since the sensitivity of MPS is mainly based on the particle core composition and not on the environmental serum, the signal change in the experiments between binding and non-binding samples is quite small. In addition, the signal as well as its change strongly depends on the MNP and the analyte concentration, which requires sophisticated sample handling and data processing to robustly detect the relevant signal changes.

Similar modalities using MNPs, such as AC susceptometry (ACS) measurements[11–14] provide an alternative technique to investigate and determine parameters of the environmental serum, e.g., rapid detection of 84 pM mimic SARS-CoV-2 in 36 s[11]. ACS measurements can cover a larger parameter space yielding more sensitive results but with longer acquisition times.

Common MPS devices work with a strong time-varying magnetic field $H_{AC}$, while ACS devices work with weak excitation fields $H_{AC}$ below 2 mT and multiple frequencies $f_{AC}$ and sometimes with additional strong offset magnetic fields $H_{DC}$ (static or with low frequency $\ll f_{AC}$).

Here we combine a strong excitation field $H_{AC}$ with a strong offset magnetic field $H_{DC}$ and expand the parameter space with COMPASS (Critical Offset Magnetic PArticle SpectroScopy) as indicated in Fig. 1. This allows extremely sensitive and robust investigations of MNP dynamics and surface chemistry at critical offset fields, which to our knowledge has not been exploited before, and allows for measurements with higher sensitivities than MPS or ACS. Furthermore, COMPASS reaches a detection limit of SARS-CoV-2 S1 antibodies binding to the S antigen on a functionalized surface of MNPs, which is comparable with the gold-standard methods ELISA (Enzyme-linked Immunosorbent Assay)[15] and flow cytometry[16]. While both ELISA and flow cytometry are limited by their inflexibility, complex handling, and long measurement times, COMPASS provides a robust and easy-to-use testing environment.

## Results

### Physical background of critical points

The magnetization of a superparamagnetic sample depends on the surrounding magnetic field $H = H_{AC} + H_{DC}$ consisting of dynamic $H_{AC}$ and static $H_{DC}$ magnetic fields. The simplest model is that of a single-domain particle, which can be seen as tiny permanent magnets. In absence of an external magnetic field, all nanoparticles of such an ensemble (sample) are statistically oriented, which causes the magnetization of the sample to be zero. Increasing the external magnetic field strength leads to more and more particles aligning along the external magnetic field resulting in an increase in the magnetization. At a specific magnetic field strength $H_{sat}$, all particles are aligned, and the

magnetization of the sample is saturated (saturation magnetization $M_{sat}$). The dependency of the sample magnetization $M$ on the external magnetic field strength $H$ can be described by the Langevin function $L(\xi)$:

$$L(\xi) = \coth(\xi) - \frac{1}{\xi} \text{ with } \xi = \frac{\mu_0 mH}{k_B T}, \qquad (1)$$

with $m$ as the magnetic moment of a particle, $\mu_0$ as the vacuum permeability, $k_B$ as the Boltzmann constant, and $T$ as temperature. The Langevin parameter $\xi$ describes the different regimes of the magnetization response: $|\xi| \ll 1$ describes the linear regime for small external magnetic fields and $|\xi| \geq 1$ describes the non-linear regime (Fig. 2a). However, it is important to note that Eq. (1) is only an approximation to real particles. Especially the assumption that the magnetization follows the external field instantaneous is not fulfilled. As a matter of fact, the presented method requires particles with strongly blocked magnetic moments, i.e., they show no or negligible Néel relaxation and therefore can only rotate mechanically (Brownian relaxation, Supplementary Note 2).

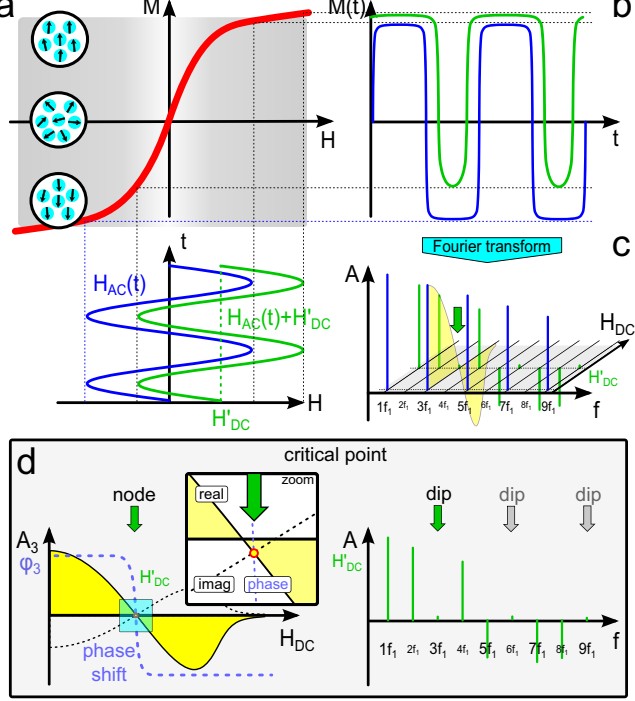

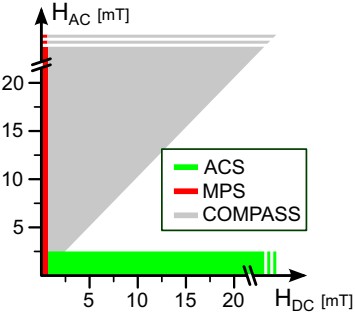

**Fig. 1 | Parameter space for different MNP measurement methods.** ACS and common MPS are using either weak AC and strong DC magnetic fields (static or with low frequency $\ll f_{AC}$) or strong AC and weak DC magnetic fields. At critical DC magnetic field offsets in the strong AC and DC magnetic field regime ($H_{DC} < H_{AC}$), the signal phase can be especially sensitive to small changes in the MNP mobility—**C**ritical **O**ffset **M**agnetic **PA**rticle **S**pectro**S**copy (**COMPASS**).

**Fig. 2 | Origin of the critical point. a, b** The behavior of the magnetization $M$ of MNPs in dependency of external magnetic fields $H$ can be described by the non-linear Langevin function (red curve). Exposing an MNP ensemble to a sinusoidal magnetic field $H_{AC}(t)$ with frequency $f_1$ and sufficient amplitude, the magnetization response $M(t)$ consists not only of the fundamental frequency but also odd (and even) higher harmonics ($f_n = n \cdot f_1$) depending on the presence of an offset magnetic field $H_{DC}$, which can be visualized in the Fourier spectrum. **c** Visualizing the dependency of the harmonic $A_n$ for the $n$-th higher harmonic for increasing offset magnetic field $H_{DC}$ (with $H_{DC} < H_{AC}$). The specific shape for varying $H_{DC}$ with nodes (green arrow) depends on the harmonic number $n$. As an example, the real part of the third harmonic of simulated data is indicated to show the connection between a "dip" in the Fourier spectrum and a "node" in the $A_n(H_{DC})$ plot: this point is called the critical point (CP). **d** In the vicinity of a CP, the position, which is most susceptible to the sample parameters, the phase $\varphi(H_{DC})$ of the signal shows an approx. 180° degree shift with a strong slope. Thus, even minimal changes in the sample parameters and thus in the shape of the $A_n(H_{DC})$ and $\varphi(H_{DC})$ curves result in high changes in the phase $\varphi(H_{DC})$ and thus in the resulting signal. The sixth and ninth harmonic are integer multiples of three and therefore happen to also vanish in this case (gray arrows).

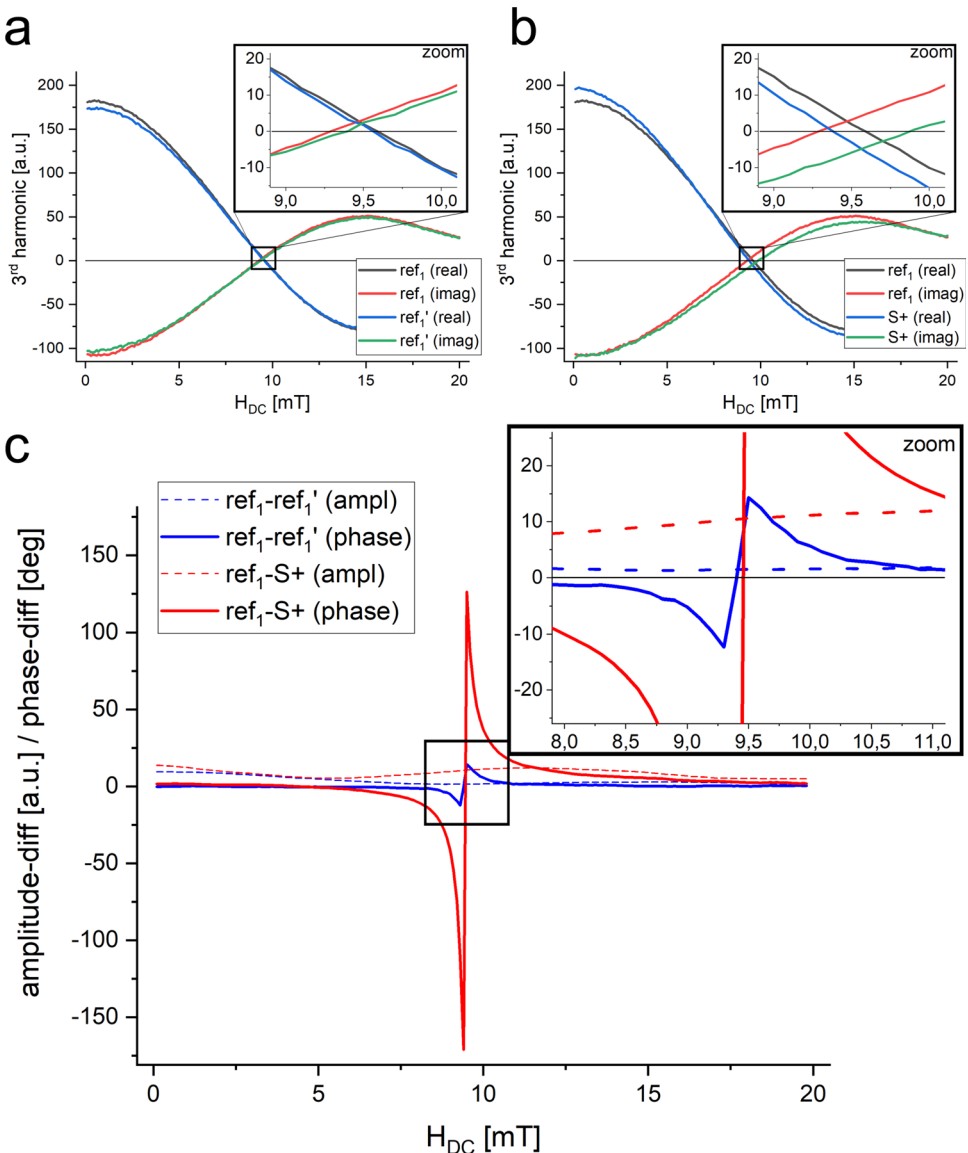

**Fig. 3 | Critical point sensitivity. a, b** The real and imaginary parts for the third harmonic ($H_{AC} = 17$ mT) of experiments with a reference sample ($ref_1$ & $ref_1'$) and a reference sample and a binding sample ($ref_1$ & $S+$). The differences in the crossing points of real and imaginary data are clearly visible. **c** The phase difference (solid lines) as well as the amplitude difference (dashed lines) of both experiments differentiate the binding sample ($ref_1$ & $S+$) from control ($ref_1$ & $ref_1'$).

MPS devices are using time-varying magnetic excitation fields $H_{AC}(t) = H_0 \cdot \sin(2\pi f_1 \cdot t)$, which are sufficiently high to drive the magnetization $M$ of a sample periodically with frequency $f_1$ into their nonlinear response. In contrast, the magnetic field strength of ACS devices is much smaller ($H_{0,ACS} < 2$ mT $< H_{0,MPS}$). That means ACS investigates the behavior of the sample in the linear regime ($|\xi| \ll 1$) determining the susceptibility or slope ($\chi = dM/dH$) of the magnetization curve while MPS is more focused on the non-linear response of the magnetization ($|\xi| \geq 1$).

The magnetization response $M(t)$ over time of a sample during continuous magnetic field excitation $H_{AC}(t)$ larger than 5 mT (MPS) approximates a mostly rectangular shape depending on the amplitude $H_0$ of the excitation field. An analysis of the time signal using a Fourier transformation reveals odd higher harmonics $(2n-1) \cdot f_1$ ($n \in \mathbb{N}$) of the excitation frequency $f_1$ in the spectrum due to the symmetric behavior of the signal over one period $1/f_1$. These higher harmonics are specific for the MNP type and encode information of its magnetic response and, hence, on the properties of the particle or its surrounding.

During ACS experiments, only the fundamental frequency $f_1$ is usually studied at different frequencies in the linear regime to get a frequency-dependent characterization of the MNPs[11-14].

For both ACS and MPS the application of static offset magnetic field $H_{DC}$ parallel to the excitation field ($H_{AC}(t) \| H_{DC}$) extends both methods and allows for a closer investigation of the magnetization curve in the non-linear regime.

During MPS experiments in the presence of an offset magnetic field $H_{DC}$ the magnetization response $M(t)$ becomes asymmetric, which introduces higher even harmonics $2n \cdot f_1$ ($n \in \mathbb{N}$) of the excitation frequency $f_1$ in the Fourier spectrum (Fig. 2a, b).

Investigating the spectral components $A_n$ of each higher harmonic $n$ in dependence of the offset magnetic field strengths $H_{DC}$, the real and imaginary part of $A_n(H_{DC})$ show an interesting behavior. For $H_{DC} < H_{AC}$ a wavelike functional dependence on $H_{DC}$ with zeroes, also called nodes, at offset fields specific for each harmonic $n$ is observed (Fig. 2c, d and Supplementary Movies 1 and 2). This behavior can be described by a convolution of Chebyshev polynomials of second kind $U_n$ with the derivative of the magnetization curve $M' = dM/dH$

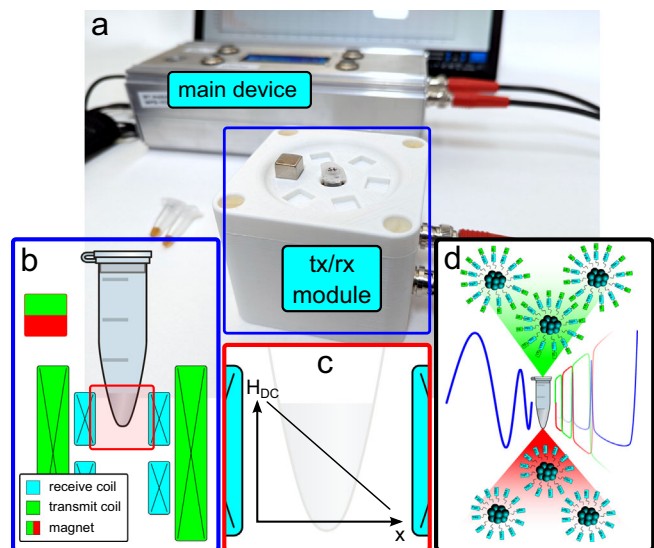

**Fig. 4 | Mobile COMPASS device. a** Shows the main device consisting of all necessary hardware components such as a microcontroller, amplifier, and filters, the transmit/receive module for 0.5 ml Eppicaps. **b** A closer look at the coil design within the tx/rx module indicates the positioning of the samples within one of the gradiometric receive coils. **c**, **d** The offset magnetic field $H_{DC}$ along the sensitive area within the rx coil shows a strong gradient $G$, which has a massive influence on the signal differentiation of binding and non-binding states of functionalized MNPs.

(Supplementary Note 1)[17]. With increasing harmonic number $n$, the spectral component $A_n(H_{DC})$ shows an increasing number of nodes. The corresponding phase plot $\varphi_n(H_{DC})$ of the harmonic signal shows a steep slope of the phase near such nodes or "dips." Hence, minimal changes in the magnetization response curve due to changes in particle or environmental parameters, e.g., hydrodynamic diameter, lead to a strong detectable phase difference $d\varphi = \varphi_{res} = \varphi_1 - \varphi_2$ between two experiments with two different samples (Supplementary Note 2). This implies a high sensitivity to changes in the sample parameters at these distinct offset field-induced nodes which are, hence, called critical points (CPs) in the following.

### Critical points sensitivity evaluation

To evaluate the COMPASS method, we hypothesized that we can exploit COMPASS to detect the binding of SARS-2 specific antibodies with sensitivities competing with ELISA and flow cytometry. With an incubation time of only a few seconds, it provides a detectable signal change of more than 10 standard deviations[18] (Supplementary Note 9), leading to a real rapid testing protocol (Supplementary Note 7).

Multiple samples with slightly different hydrodynamic diameters were prepared and measured in vitro with the aim of detecting commercially available SARS-CoV-2-specific antibodies. For the binding sample (S+), SARS-CoV-2-S1 protein was covalently bound to the surface of MNPs functionalized with (3-Aminopropyl)tiethoxysilan (APTES) using a protocol modified from ref. 19 and resulting in MNP-APTES-S1. The preparation of the samples for the measurements was the following (Supplementary S3): antibodies were diluted 1:2000–200,000 (3.3–33 pM) in a buffer (PBS with 0.1% BSA). For each measurement, 25 μl antibody dilution or reference sample (dilution buffer) was added to 25 μl of MNP-APTES-S1 dispersions (100 μg Fe/ml) in an 0.5 ml Eppendorf cap. After adding the antibody dilution or buffer (reference sample), the samples were mixed shortly by pipetting and directly measured without any further incubation time. The reference sample (ref) contained the MNP-APTES-S1 and a buffer solution.

In Fig. 3a, b, the real and imaginary parts of the third harmonic of two experiments with two samples each, in dependency of a stepwise

increased offset magnetic field $H_{DC}$ are shown. These "full data sets" were acquired with a benchtop MPS device with an adjustable offset magnetic field system. The amount of single data sets includes 7200 single measurements per sample and required several minutes of acquisition time each (Supplementary S4).

A closer look at the nodes of each experiment revealed differences between the crossing points of the samples. The reference-vs-reference measurement ($ref_1$ & $ref_1'$, Fig. 3a) showed almost no difference in the signal demonstrating the stability of the measurement. In contrast, the difference between a reference and a binding-sample ($ref_1$ & $S+$, Fig. 3b) while subtle was clearly detectable. In Fig. 3c, the differences between both experiments are indicated (amplitude differences and phase differences). Two prominent results became evident: first, the difference in the peak height and width of the phase differences (solid line) between both experiments. The phase difference between a reference and binding sample measurement is by a factor $f_{d\varphi} \approx 17$ increased compared to the phase difference of two reference samples. Second, the height of the amplitude difference (dashed lines), especially in the range of the peak, differed strongly and approaches zero for the $ref_1$ & $ref_1'$ measurement. Including this calculated amplitude difference factor $f_{dA}$ of about 10 would also help to distinguish noise from true signal changes in the vicinity of the critical points.

The phase difference for the $ref_1$ & $ref_1'$ measurement also showed a clearly visible peak, which lay, as expected, at the highest phase sensitivity of the system (critical point). This effect is dominated by noise and slightly by systematic errors such as sample positioning between the successively performed measurements. This reflects an intrinsic sensitivity limit of the used device.

The initial result in Fig. 3 revealed not only a high sensitivity on minimal changes of particle diameters (mobility) in the vicinity of each CP for each higher harmonic but also indicated a high robustness on hardware requirements or magnetic field parameters due to the width of the peak.

### Mobile COMPASS device

Many measuring techniques are based on physical effects and their sensitivity commonly correlates with the complexity of the underlying measurement hardware. With increasing demand on sensitivity, the requirements for sophisticated hardware to guarantee the necessary specificity and robustness increase significantly. Thus, depending on the desired application, such methods may become non-feasible.

The observed results suggest design parameters and design specifications for a highly flexible and robust device allowing very sensitive and specific measurements. The device presented in the following is based on common MPS technology running at a base-frequency $f_1 = 20$ kHz and comes with a robust hardware design and easy-to-handle experiments[6]. Based on the results shown above, an important hardware modification was introduced by adding a strong permanent magnet, which generates a strong magnetic field gradient $G$ along the measurement area providing a range of offset magnetic fields $H_{DC}$ within the sample volume. Under the right condition between excitation field $H_{AC}$ and offset magnetic fields covering one or more CPs, the sensitivity of MPS experiments against minimal changes in mobility was improved significantly (Supplementary Note 2).

In Fig. 4a, the proposed modified MPS device is shown. As a mobile and highly flexible stand-alone device, it consists of the main control device with all required electronic parts, such as transmit/receive (tx/rx) module for generating the required magnetic fields and measuring the sample signals as well as a battery pack as power supply. The sketch in Fig. 4b of the tx/rx module shows a cross-section through the tx/rx module indicating the position of the sample in the sensitive area in the center of one of the receive coil pair (rx) wired as gradiometer, the transmit solenoid (tx) and the permanent magnet. The offset magnetic field $H_{DC}(x)$ generated by the permanent magnet creates a strong magnetic field gradient $G$ within the measurements

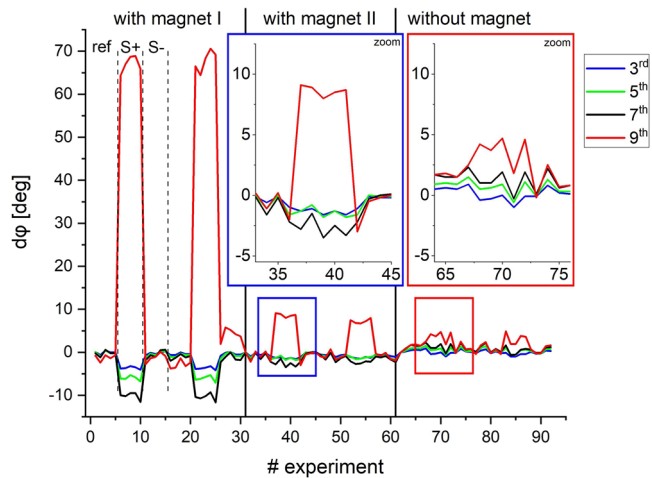

**Fig. 5 | First measurements with mobile modified MPS device on binding states on MNP-APTES-S1.** Comparison of signals with and without offset magnetic fields (gradient field). All sequences show 5 experiments with reference sample (*ref*), 5 with binding sample (*S+*) and 5 with non-binding sample (*S−*) repeated two times. **With magnet I** The signal (phase difference) is clearly visible. **With magnet II** The rotated magnet yields a lower magnetic field resulting in a signal loss. **Without magnet** The signal almost vanishes without any external magnetic offset or magnetic gradient field. For all experiments, nanoparticles ligated with the SPIKE (S1) protein of the SARS-CoV-2 virus with an antibody dilution of 1:10,000 was used.

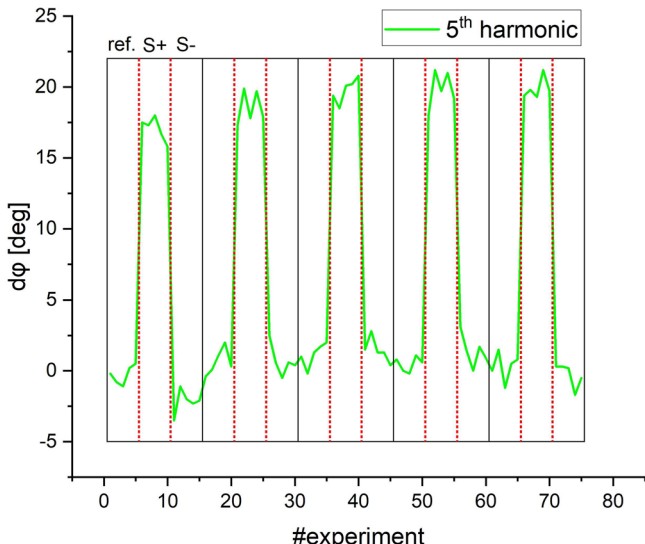

**Fig. 6 | High-sensitive measurement with COMPASS.** Results of the measurements with an optimized setup (Supplementary Note 5) on the 5th harmonic of the binding states of a dilution of 1:500,000 SARS-CoV S antibodies (corresponding to 0.33 fmol per 50 μl sample volume or ~7 pM) on MNP-APTES-S1. The measurement sequence was 5 times reference sample (*ref*), 5 times binding sample (*S+*), and 5 times non-binding sample (*S−*) and was repeated five times resulting in 75 individual experiments. Note that no visible difference between reference (*ref*) and non-binding sample (*S−*) is visible. The acquisition time was 10 ms per experiment with a repetition time of 1 s. All samples have been prepared only seconds before measurement.

chamber (Supplementary Note 5), which influences the inductively measured signal significantly.

In Fig. 5a, the first results of the proposed mobile-modified MPS device measuring the binding state of MNP-APTES-S1 particles are shown (for data processing details, see Supplementary Note 6). Each measurement was performed 5 times without averaging. The sequence of samples was reference sample (*ref*) containing buffer, binding sample (*S+*) containing a S1 binding antibody (SARS-CoV-2-S1 antibody) and non-binding-sample (*S−*) containing a non-binding antibody (MERS-CoV-S1 antibody) and was repeated 2 times resulting in 30 individual measurements. The acquisition time for each experiment was 10 ms with a minimum repetition time of 1 s. The graph shows the phase difference $d\varphi_n$ on selected higher harmonic ($n = 3$rd to 9th) against the reference sample. A significant phase difference on each harmonic was observed for the binding sample (*S+*) but not for the non-binding sample (*S−*). Here, the ninth harmonic showed the highest difference, but also other harmonics revealed significant phase differences since the applied gradient (permanent magnet) ensured a broad range of offset magnetic fields acquiring signals from multiple critical points.

For comparison, a series of experiments were performed to demonstrate the influence of the magnetic offset fields and magnetic field gradient on the signal (Fig. 5b). For that, the same experiment sequence was performed for three different cases: (1) with a permanent magnet in described position, (2) with permanent magnet in a rotated position (90° degrees against the tx/rx orientation) and (3) without permanent magnet. It became evident, that in case (1) and (2) (with permanent magnets) the desired signal (phase difference) was more prominent than without (case (3)). The signals with permanent magnets differed depending on the gradient strength generated by the permanent magnet within the measurement chamber. This variation depended on the range of offset fields (gradient G) mentioned above.

### Can COMPASS be an alternative to ELISA and flow cytometry?
The results in Fig. 5 with the modified MPS device represent the signal not only at one specific position $H_{DC}$ of the Chebyshev-like polynomial (c.f. Fig. 3) but the integration of signals over a range of offset magnetic

fields (Supplementary Note 2). The sensitivity strongly depends on the chosen gradient field G as indicated in Supplementary Fig. 4. As shown in Fig. 3, the sensitivity of the method for specific harmonics increased further by adjusting the gradient field around a very small and specific range covering the area of a specific critical point $CP_{i,j}$ (Supplementary Fig. 12) but potentially at the cost of reduced robustness.

However, despite the simple setup, the sensitivity of the mobile COMPASS device reaches ~2 ng/ml (0.33 fmole per 50 μl-sample volume) of SARS-CoV-2-S1 IgG antibody (see Fig. 6), which is comparable to the sensitivity of flow cytometry devices (100–200 ng/ml) as well as the sensitivity of ELISA tests (20–40 ng/ml)[15] (Supplementary Note 8).

In addition, experiments with blood serum with different amounts of antibody concentrations have been performed to demonstrate the robustness in a more realistic environment. For that, in an initial calibration experiment, a negative blood serum (serum0−) is used to investigate possible cross-binding effects. As indicated in Fig. 7 top, only the third sample, consisting of MNP-APTES-S1 + serum0− + SARS-CoV-2 S1 antibodies, shows a clear signal, whereas the other samples, MNP-APTES-S1 + serum0− and MNP-APTES-S1 + serum0− + MERS antibodies, show the same signal as the reference (pure MNP-APTES-S1) demonstrating the absence of unspecific binding for these controls.

In Fig. 7 bottom, a small study is provided, showing the COMPASS results of multiple samples with different amounts of anti-spike IgG in blood serum (high-level serum1+ with 28,600 BAU/ml, medium-level serum2+ with 6500 BAU/ml, and low-level serum3+ with 44 BAU/ml). Serum anti-spike IgG levels were quantified using the LIAISON® SARS-CoV-2 TrimericS IgG assay (cut-off: 34 BAU/ml, time to first results ~35 min).

Clearly, two effects can be seen, first, there is no cross-linking visible in the calibration experiment (Fig. 7a) and second, a clear signal can be obtained for the serumX− probes even for low-level sample (Fig. 7b).

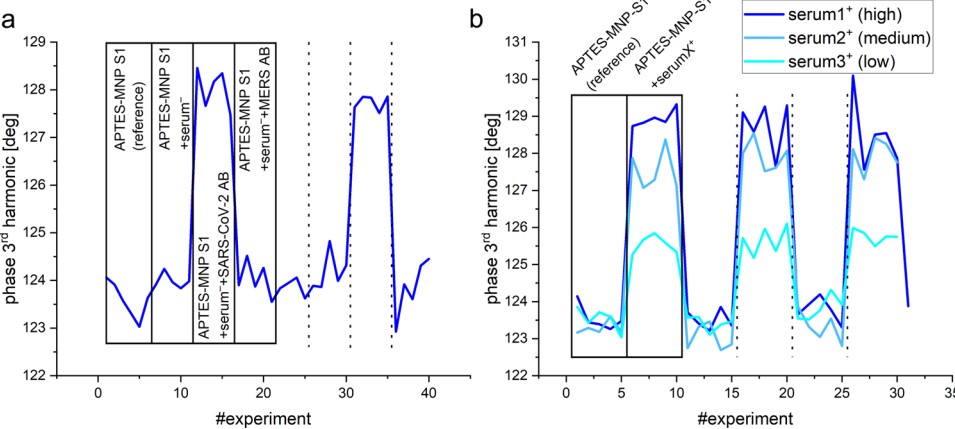

**Fig. 7 | First blood serum experiments. a** Calibration experiment investigating possible cross-binding effects between blood serum and APTES-MNP-S1 particles. The measurement sequence was 5× reference, 5× serum0−, 5× serum0−+SARS-CoV-2 ABs, 5× serum0−+MERS ABs with two repetitions for calibration experiment. **b** Results for three different blood sera obtained from healthy staff members of the Institute of Virology and Immunobiology with informed consent. Serum anti-spike IgG levels were quantified using the LIAISON® SARS-CoV-2 TrimericS IgG assay (cut-off: 34 BAU/ml) for a high level (serum1+ = 28,600 BAU/ml), medium level (serum2+ = 6500 BAU/ml), and low-level sample (serum3+ = 44 BAU/ml). The measurement sequence was 5× reference, 5× serumX+ with three repetitions for serum measurements (X means the different conc. level) to demonstrate measurement reproducibility. The acquisition time was 10 ms per experiment with a repetition time of 1 s. All samples have been prepared only seconds before measurement.

## Discussion

Importantly, the handling of COMPASS experiments and measurements is flexible and requires no complicated sample preparation and the results are robustly available in a short protocol time (only a few seconds including mixing and incubation time) as indicated in Fig. 6. However, in a more realistic case using blood serum consisting of different amount of SARS-CoV-2 antibodies (Supplementary Note 9), additional serum preparation time is required (~10 min)[20]. Furthermore, a quantification of the amount of bindings on the surface of the functionalized particles was observed with COMPASS at a high specificity (Supplementary Note 8). Our method can be used as a robust, fast and easy-to-handle and cheap testing method for sensitive and specific antigen or antibody determination. It thus offers a wide variety of applications in clinical chemistry and biomedical analytics. COMPASS also allows the measurement of intermolecular interactions of different compartments on functionalized magnetic particles. This opens a wide field in physics, medicine, biology, and chemistry[10,21,22].

Since the basic effect of the presented COMPASS method is based on a magnetic offset field-induced effect, which occurs for excitation magnetic fields $H_{AC}(t)$ as well as offset magnetic fields $H_{DC}$ with magnitudes of at least 2–3 mT or above, this technique differs from common MPS and ACS experiments (Supplementary Note 4). Setting up a critical point by adjusting the AC and DC magnetic fields enables the measurement of minimal changes in the effective mobility of the samples. The high sensitivity at the critical point is caused by a kind of background suppression, where the background can be defined as a signal from particles unaltered in their mobility in the presence of the analyte.

In addition, the differential measurement (sample-vs-reference) of the phases can overcome issues in signal interpretation occurring in MPS or ACS experiments due to concentration dependencies. This provides a huge list of particle parameters accessible with high accuracy, which can be seen in the Langevin equation (c.f. Supplementary Equation 3) such as the magnetic moment of the particle $m$, the Temperature $T$, and the friction $\zeta$, where the latter is the product of viscosity $\eta$ of the surrounding medium, the hydrodynamic particle radius $R_H$ and the particle shape $\kappa$. The advantage of this method is the direct access to particle parameters, which are of high interest for understanding the complex dynamics of MNP ensembles. Furthermore, fast and easy access to these parameters allows a robust MNP characterization during synthetization and hence gives immediate feedback for improving the quality of magnetic particles, e.g., for medical applications or environmental treatment[23,24]. Many applications in different fields of research are conceivable, and COMPASS could pave the way for their realization.

## Methods

### Ethical statement
Serum samples were obtained from lab members with informed consent. This study was approved by the ethics committee of University Hospital Würzburg (AZ 35/07).

**Measurement head.** The measurement head of the COMPASS device consists of a 3D-printed box that contains the transmit- and receive coils (height: 64 mm, width: 80 mm, depth: 80 mm). The sample is placed on top of the box inside the receive coil opening. The transmit coil is placed around the receive coil and axially aligned. The transmit coil has 36 windings around an inner cross-section of 9 mm with litz wire with 90 strands of 100 μm each, resulting in a resistance of 70 mΩ and a magnetic field of 2.1 mT/A. The receive coil is wound with litz wire with 12 strands of 40 μm each around an inner cross-section of 5 mm. It consists of two coils with 20 windings each and with opposite winding direction for suppressing the excitation signal, resulting in a resistance of about 2 Ω. The receive coil can fit 0.5 ml Eppicaps. A magnetic offset field with a gradient is generated using a Neodym N52 permanent magnet. The magnet is placed on top of the Box (30 mm above the center of the sample position and 20 mm apart from the receive coil axis). The axis through the magnetic poles is aligned in parallel to the receive coil axis. For more information see Supplementary Note 5.

**Transmit chain.** The transmit coil is driven with a linear audio amplifier chip (TDA7294). It is connected with the transmit coil with a simple capacitive match-and-tune network (match: 666 nF, tune: 3533 nF). The transmit coil is driven with a current of about up to 25 A at 20 kHz, generating about 50 mT. For more information, see Supplementary Note 5.

**Receive chain.** The receive chain is matched to a low-noise amplifier with a transformer and a 5-pole bandpass filter. The signal conditioning for matching the dynamic range of the ADC was done using an on-chip

programmable gain amplifier. For more information, see Supplementary Note 5.

**Controlling device.** Both the transmit and receive amplifier are mounted on a PCB Board in an aluminum box (length: 220 mm, width: 105 mm, height: 70 mm). A microcontroller from Cypress Semiconductors is used to generate the 10 ms long sequence (on-chip DAC: 1 MS/s with 8 bit), to acquire the received signal (on-chip ADCs: 2×1 MS/s with 12 bit, combined for achieving 2 MS/s) and to perform all necessary data processing. The microcontroller was programmed using PSoC™ Creator. Results are displayed on an LCD module. The phase value is displayed in degrees for on higher harmonic at a time (selected via bush button). The raw data are also accessible via an UART-to-Bluetooth module, which allows separate data processing with a mobile device. For more information, see Supplementary Note 5.

**Sample preparation.** For the detection of SARS-CoV-2-S1 antibodies using the COMPASS device, the stock solution was diluted in buffer to 1:2000, 1:5000 1:10,000, 1:20,000 and 1:200,000 (corresponding to 5 ng/ml for the lowest antibody concentration). An amount of 25 µl of MNP-APTES-S1 dispersions (100 µg Fe/ml) was added in an 0.5 ml Eppendorf cap. Subsequently, 25 µl of antibody dilution (S+) or buffer (*ref*) were added. Samples were directly measured in the COMPASS device after careful mixing by pipetting without any further incubation times. For more information, see Supplementary Note 3.

**Measurement protocol.** The measurement protocol for testing with COMPASS is optimized for receiving clear and robust results rapidly within seconds. For that, a differential measurement of two identical probes split from the original prepared MNP-APTES-S1 batch is used. The to-be-measured substance is directly given in the sample probe and can be instantaneously measured with COMPASS without further washing processes or conjugation or incubation times. The same amount of buffer solution is added to the reference sample before being measured with COMPASS. The results are available within seconds. For more information, see Supplementary Note 7.

**Binding kinetics.** For measuring the real-time binding kinetics of APTES-MNP-SBA-S1 + MERS antibodies (neg. control) and SARS antibodies, a time series measurement with one measurement every second was performed using COMPASS. 20 seconds after start of the measurement series, 25 µl MERS antibodies (dilution 1:50k−20 ng/ml) were added. After 110 s, 25 µl SARS antibodies (dilution 1:50k−20 ng/ml) were added. For more information, see Supplementary Note 9.

**MNP preparation.** MNPs functionalized with APTES ((3-aminopropyl) triethoxysilane, Carl Roth, Germany) were used as an exemplary particle system. These MNPs are multicore particles whose crystallites show an average diameter of about 12 nm. They are coated with APTES and produced by alkaline precipitation (Supplementary Fig. 5)[25]. The MNPs produced in this way have a hydrodynamic diameter of about 200 nm with the single crystallites showing a diameter of about 12 nm. SARS-CoV-2-S1 protein (SARS-CoV-2 (2019-nCoV) spike S1-His, Sino Biological, China) is covalently bound to the surface of the particles by binding SBA (N-succinimidyl bromoacetate) over cysteines present in the protein. A 0.05 M borate buffer with pH 8.5 was used during binding. The particle concentration during the functionalization processing was adjusted to 1 mg Fe/ml and 20 mM SBA dissolved in DMF (Carl Roth, Germany) was added. The samples were shaken for 2 h at 1,400 rpm. After that, the particles were washed several times with a buffer solution. The obtained MNP-APTES-SBA were redispersed in borate buffer for binding of SARS-CoV-2-S1 protein. The samples were shaken again for 2 h at 1400 rpm and washed several times with doubly distilled H₂O. The final MNP-APTES-SBA-S1 possesses a concentration

of 10 µg S1 protein per 100 µg Fe (determined by UV-VIS measurements). After binding and the last washing step, the hydrodynamic size of the multicore particles (MNP-APTES-S1) is around 330 nm with a PDI of 0.3. Particles were stored in doubly distilled water until further use. For more information, see Supplementary Note 3.

**Flow cytometry analysis.** To test the binding selectivity to the correct antibody flow cytometry-analysis was performed with a Gallios flow cytometer (Beckman Coulter, Fullerton, CA, USA). First the antibodies (conc. 1 µg/ml SARS-CoV/SARS-CoV-2 Spike antibody (S1 antibodies, MW: 146.16 kDa), Chimeric Mab, Sino Biological, China) were diluted 1:2000, 1:5000, 1:10,000, 1:20,000 (last dilution corresponding to 50 ng/ml) antibodies in a buffer (PBS with 0.1% BSA). For the measurement 25 µl of MNP-APTES-S1 dispersions (iron conc. 100 µg Fe/ml) were added in an 0.5 ml Eppendorf cap. Subsequently 25 µl of antibody dilution or buffer (*ref*) were added. Samples were incubated for 1 h at 4 °C. For washing, samples were centrifuged at 18,000 rcf for 10 min, the supernatant discarded and the MNPs redispersed in buffer. For detection, MNPs were further incubated with fluoresceinisothiocyanate (FITC) labeled protein A (1 µg/ml) for 1 h at room temperature (RT), which is known to bind specifically the Fc-region of IgG-antibodies[26]. Finally, the MNPs were washed as described before, redispersed and diluted 1:250 in buffer, and analyzed for fluorescence by using flow cytometry. The fluorescence bleed through was eliminated by electronic compensation. The acquired data were analyzed with Kaluza software version 2.0 (Gallios, Brea, USA). For more information, see Supplementary Note 8.

**ELISA.** For human CoV-19 ELISA for S1 antibodies (RayBio, Peachtree Corners, GA), the S1 antibodies were diluted from 1500 ng/ml to 25 ng/ml in the same buffer as used for the other tests. For the ELISA an amount of 100 µl sample were added to each well. After 1 h of incubation at RT under gentle shaking, all liquid was depleted, and the plate was washed four times with washing buffer. 100 µl of prepared Biotinylated Anti-Human IgG Antibody is added to each well followed by incubation for 1 h as carried out before followed by another washing as described. Next 100 µl of prepared HRP-Streptavidin solution was pipetted into each well, followed by another incubation for 30 min. After washing 100 µl of TMB One-Step Substrate Reagent was applied to the wells. After a further 30 min of incubation, 50 µl of Stop Solution was added. The OD at 450 nm was detected using a plate reader (SpectraMax iD3, Molecular Devices). For more information, see Supplementary Note 8.

**Parameter space evaluation.** To get a better understanding of the signal behavior at the critical points and the behavior of the magnetic field configurations, a more sophisticated experiment was performed to obtain a full data set covering the full range of AC fields and DC fields from 0 to 20 mT. For that, the samples were measured with multiple offset magnetic fields $H_{DC}$ as well as multiple excitation magnetic field strengths $H_{AC,j}(t)$ to obtain a data set over a defined range of both parameters. For that, an additional large solenoid with a diameter of 30 cm has been placed around the AC-field generator box. The DC values were adjusted step-by-step by a computer-controlled DC power supply. The result is a 2D contour plot for real and imaginary parts of the MNP signal depending on the offset magnetic field $H_{DC}$ and excitation magnetic field $H_{AC,j}(t)$ for each harmonic $n$. The range of the offset magnetic field spans 0 mT up to 20 mT in 0.1 mT steps. The range of the excitation magnetic field spans 0.6 mT up to 21.6 mT in 0.3 mT steps. In sum, 200·36 = 7200 individual measurements were performed for each full data set. The basic excitation frequency was $f_1 = 20$ kHz. For more information, see Supplementary Note 4.

**Simulations.** For theoretically modeling COMPASS measurements, a basic simulation using the Langevin equation was devised[27,28].

Simulated were two different particle systems with a minimal difference of about 19% in their viscosity, which corresponds to, e.g., a small change in the effective hydrodynamic particle diameter of about 6%, were performed. This was done using a simulation framework developed at Würzburg for calculating magnetic fields and non-linear magnetization responses on time-varying magnetic fields[29]. The parameters used in the Langevin equation were the magnetic moment $m = 4 \cdot 10^{-16}$ A·m$^2$, the viscosity $\eta = 1$ mPa·s, the shape factor $\kappa = 3$ (spherical shaped particles), and the hydrodynamic diameter $R = 330$ nm and the temperature $T = 300$ K. For more information, see Supplementary Note 2.

### Reporting summary

Further information on research design is available in the Nature Portfolio Reporting Summary linked to this article.

### Data availability

Raw data, preprocessed measurement data, as well as specific source files (Inkscape V1.2, Origin 2021b) for generating relevant graphs are available on zenodo.org (https://doi.org/10.5281/zenodo. 7304376). Source data are provided with this paper.

### Code availability

Source codes are provided with this paper. Source code of user-defined data analysis software (Embarcadero RAD Studio 11) for data visualization and processing is provided on zenodo.org (https://doi. org/10.5281/zenodo.7304376).

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

### Acknowledgements

Research funding: The work was supported by the German Research Council (DFG) (grant numbers: VO-2288/1-1 [P.V.], VO-2288/3-1 [P.V.], BE 5293/1-2 [V.C.B.]), the Manfred Roth Stiftung, Fürth, Germany and the "Forschungsstiftung Medizin am Universitätsklinikum Erlangen," Erlangen, Germany. This publication was supported by the Open Access Publication Fund of the University of Wuerzburg.

### Author contributions

P.V.: initial idea, hardware development, assembling spectroscope, performing experiments, preparing figures, data processing, provides

software for data processing and simulations, writing the manuscript. M.A.R.: initial idea, hardware developing, assembling spectroscope, evaluating theory, writing the manuscript B.F.: sample preparation, experimental design, performing experiments, data processing, writing the manuscript R.T.: initial idea, sample preparation, experimental design, writing the manuscript S.L.: initial idea, sample preparation, experimental design, writing the manuscript T.K.: evaluating theory, writing the manuscript T.H.: experimental design L.D.: initial idea, experimental design, writing the manuscript C.A.: providing chemistry and biology labs, resource management V.C.B.: providing MPI lab, resource management, writing the manuscript. All authors reviewed the manuscript.

## Funding

## Competing interests

P.V., M.A.R., B.F., R.T., S.L., and C.A. have submitted a patent application to BayPAT pertaining to the COMPASS method (application number EP22159238). The remaining authors declare no competing interests.
