## [Peer Review File · Nature Communications]

COMPASS – rapid and highly sensitive medical point-of-care diagnosticEditorial Note: Parts of this Peer Review File have been redacted as indicated to remove third-party material where no permission to publish could be obtained.

REVIEWER COMMENTS

Reviewer #1 (Remarks to the Author):

This manuscript proposes a novel method called COMPASS, where a critical offset DC field is applied together with an AC excitation field in a low-cost Magnetic Particle Spectroscopy (MPS) device to elicit highly sensitive measurements of subtle changes in the hydrodynamic diameter of Magnetic Nanoparticle (MNP) ensembles. The authors also propose adding strong magnetic fields to further increase the sensitivity to detect 33 pM of SARS-COV-2-S1 antibodies, already demonstrating higher sensitivity than gold standard methods ELISA and flow cytometry, with the added benefit of much shorter protocol times (under a minute). While MPS was previously utilized to detect changes in mobility, those methods mostly relied on detecting the changes in the overall signal level. The novelty of the proposed COMPASS method lies in the fact that it does not rely on global signal changes that may be sensitive to factors such as concentration differences with respect to a reference, but rather takes advantage of abrupt phase changes in the signal at predetermined higher harmonics.

The COMPASS method is highly novel and extremely practical at the same time. It will surely have a high impact in point-of-care diagnostics and lead the way to numerous new applications. The manuscript is well written in general. My main comments are listed below, and the other points are marked in the attached PDF files (main manuscript and supplementary).

Main Comments:

- Please comment on how/whether the optimal operating parameters of the COMPASS device depend on MNP parameters and the binding vs. non-binding hydrodynamic diameter.
- The iron concentration in 25 uL of MNP-APSTES-S1 dispersion was reported as 100 ug Fe/mL, corresponding to about 2.5 ug of Fe. This is a relatively decent amount of iron, which must help provide a decent SNR for COMPASS measurements, further increasing its robustness. Please comment on whether this is a realistic amount of iron, or whether real-life point-of-care diagnostics would require the samples to be prepared at smaller volumes and/or smaller concentrations.
- Supp. S2: "minimal difference of about 5% in their ζ -parameter". Is it known how different the hydrodynamic diameter is for the binding sample vs. the reference sample in the experiments? Is it more than 5%?
- Results in Fig. S2-1 and Fig. S2-2: These simulation results are dependent on the MNP parameters and H_{AC} parameters. For repeatability purposes, important parameters of this simulation should be provided.
- The simulations utilized offset DC fields in the same axis as the AC field, whereas Fig. S5-2 shows that the offset DC field from the permanent magnets is at an angle to the AC field. The authors then test two different configurations for the magnet in Fig. 5b, but neither of these cases coincide with the simulated case. Please comment on whether it is better for the DC field to be parallel or orthogonal to the AC field.
- Supp. S4: The authors mention that combining phase and amplitude can provide further increase in the difference between the binding vs. non-binding samples. In this work, the reported sensitivity of 33 pM for COMPASS was based on the phase only. I understand that this manuscript presents the general framework of COMPASS, already providing extensive analysis on numerous parameters, which is highly appreciated. Is there a reason the amplitude effect was not incorporated in this work? Are there any disadvantages to incorporating the amplitude (e.g., potential sensitivity to concentration variations)? Please comment.
- Supp. S5: Calibration steps for COMPASS do not need any measurements with a binding sample.

This sounds like a huge advantage and should be further emphasized. With that said, would the calibration improve if it incorporated measurements with both a reference sample and a binding sample (e.g., at high AB concentration)? Please comment.

Emine Ulku Saritas

Reviewer #2 (Remarks to the Author):

Several approaches to realize highly sensitive homogeneous bioassays based on functionalized magnetic nanoparticles (MNP) as markers have been reported for the detection of proteins, antibodies etc. Generally, the change of the Brownian relaxation time caused by an increase of the hydrodynamic particle size upon binding of analyte is detected. This includes measurements of the complex (AC) susceptibility and magnetic particle spectroscopy (MPS). In this manuscript, the authors describe an MPS-based approach. In contrast to MPS-based assays by other groups, the authors apply an additional dc offset magnetic field parallel to the ac magnetic field. As a certain strength of this offset field, the amplitude of a given harmonic changes sign and the phase exhibits an abrupt change by 180° . The authors call this field "critical offset magnetic field". Importantly and as simulations show, this critical offset magnetic field is very sensitive to changes of the hydrodynamic particle diameter, which increases upon conjugation of analyte, or medium viscosity, proposed that the MNP are thermally blocked, i.e. the Brownian mechanism dominates the dynamics. Thus, in the vicinity of the critical offset field, tiny changes of the hydrodynamic diameter cause a measurable change in phase. On the other hand, the signal amplitude at this critical offset magnetic field gets rather small, which will be even more critical if higher harmonics are considered as presented in the submitted manuscript. Although the approach is quite original, the manuscript is not suited for publication in Nature Communications. The manuscript just presents a novel approach for the realization of homogeneous bioassays based on functionalized magnetic nanoparticles but omits/ignores many details that are relevant for a practical application of the presented assay and that are required to justify the publication in such a high-ranking journal.

1) The authors employ comparably large MNP (hydrodynamic diameter of 330 nm after functionalization) for the detection of rather small analytes (antibodies). Using standard expression for the Brownian relaxation time and assuming room temperature and the viscosity of water, this diameter corresponds to a characteristic frequency of the nanoparticles of about 12 Hz. For comparison, the excitation frequency amounts to 20 kHz, i.e. MNP can barely follow the excitation field. This means that the system is far from being optimized and puts the presented data into question.

2) The authors claim that their approach is rapid with a measurement time of well below a minute (see abstract). In S8 of the SI, they even claim that the results are available with seconds. This is very questionable since in other ACS or MPS based homogeneous assays, which rely on the same basic principle, incubation times of up to hours were reported. The content of the manuscript does not consider any refined techniques to speed up the diffusion-limited binding kinetics. To experimentally demonstrate that the binding has really finished after a minute (or a few), measurements of the binding kinetics need to be added.

3) The reported limit of detection for the SARS-CoV-2 S1 antibodies is in the same range as the values reported by other groups (however, for spike or N proteins as well as for mimic viruses). Thus, there is no significant progress, which is needed to make magnetic nanoparticle-based bioassays compatible or superior to standard/existing techniques.

4) In Fig. S8 the authors suggest that the presented approach is readily usable for measurements in different media as e.g. saliva (nasal/throat swab) or blood. Experts in the field know that such a statement has to be considered with care since the chemistry in real media is very complex and produces many new problems. Thus, experimental data on real samples should be presented. Otherwise, the statement that the approach is universally usable should be omitted.

- 5) Authors do not explicitly emphasize that the binding scheme is based on Brownian dominated nanoparticles.
- 6) Figure such as Fig. 6 should also be shown for the S- sample to quantify the amount of unspecific binding. This is hardly discernible from Fig. 5. Unspecific binding is a major issue in such assays and needs to be studied in more detail. In addition, it should be quoted what sample was taken for the data in Fig. 5.
- 7) In the abstract and later in the text, the authors quote the sensitivity with 0.85 fmole/50 μ l sample volume, which is claimed to correspond to 33 pM. Simply taking the former numbers, one gets 17 pM. This factor of two difference needs to be explained.
- 8) One critical issue is that operating the system at the critical offset magnetic field increases on one hand the sensitivity of the phase to tiny changes of the hydrodynamic size, but on the other hand, the amplitudes of the harmonics get very small. Thus, the signal-to-noise may get too small. This point, which gets even more dramatic if higher harmonics are considered, is important since the detection of tiny amounts of analyte requires very small MNP concentrations. But such information is completely missing.
- 9) p. 1, right column: The fact that ACS is complicated by long acquisition times, is only partly true. ACS can also be speeded up if measurements are performed at a single, optimum frequency (see susceptibility reduction technique by the Horng group in Taiwan). Also, the claim that ACS requires experienced personal is not correct. Performing an ACS measurements is not more complicated than carrying out MPS measurements.
- 10) S1 and S2 in SI: Here the authors describe the modelling of the proposed assay. First, it would be important to add particle and measurement parameters that they applied in the simulation. The only explicit information they provide is that the hydrodynamics diameter is varied by 1.7%. Second, when discussing Brownian and Neel relaxation in S2, the authors quote a threshold diameter for the transition from Brownian to Neel relaxation of 30 nm. This critical diameter depends on various parameters, such as medium viscosity, shell thickness, temperature, effective anisotropy constant, and magnetic field amplitude.

Reviewer #3 (Remarks to the Author):

In this manuscript, the authors reported the COMPASS method for the detection of SARS-CoV-2 S1 antibodies. This COMPASS method circumvented the limits of traditional ACS and MPS methods that also exploit the magnetization responses of MNPs to minimal changes in the mobility of MNP assemblies. The authors have explained the mechanism of 1DC+1AC magnetic field-based COMPASS method in detail. There are some concerns that the authors need to address before further consideration for publication.

1. Critical point by its nature is extremely sensitive to the applied DC field. In that case how precision in permanent magnet placement is being taken care of? Any small deviation will result in drastic change to the results as pointed in Fig.2.
2. On the same note, information should also be pointed if the DC offset required for creation of higher harmonics differ from one another (i.e. 5th, 7th, 9th from 3rd), and if so, what harmonic has the device been optimized for concerning bioassay applications?
3. In different figures, the phase modifications of different higher harmonics are being used (refer to Fig. 5, 6, and S7-2). Considering from the end-application perspective, what is the decision-making strategy:
 - a. Considering phase change of one particular harmonic? If so than what is the harmonic of interest?
 - b. Considering a phase change in all captured higher harmonics (till 11th) and looking for a phase modification in any and all of them?
4. For the results depicted in Fig. 5, What dilution/ concentration of antibodies does the test correspond to? Also, the corresponding MNP concentration should also be noted.
5. For the reported magnetic field in the range of 17mT, with the coil parameters given, the device would have been utilizing a current of roughly 8A. Is heating of coils and hence the MNP sample present within a concern with regard to the setup? If so, what necessary steps are included to negate

this impact?

6. There is one error in the annotation in this sentence: "At a specific magnetic field strength M_{sat} , all particles are aligned, and the magnetization of the sample is saturated (saturation magnetization M_{sat})". The specific magnetic field strength is H_{sat} not M_{sat} .

7. The experiment condition is not described in detail. For example, "After adding the antibody dilution or buffer (reference sample), the samples were mixed shortly by pipetting and directly measured without any further incubation time". It's not clear how much time is waited during the pipetting step before the measurement. The authors should at least provide the estimated time.

8. A followup question from my last comment. Typically, for antibody-antigen interaction, it takes dozens of minutes up to an hour for the bindings to reach equilibrium. If the mixing time is too short, I doubt the bindings are not complete yet. Even a small mixing time difference between samples to samples will result in different binding stages. Imagine the "S" shape binding curve, if the mixing time is too short and at the steep slope of the "S" shape curve, then even a small difference in the mixing time can cause a big difference in the binding results. So, I feel the experiment design here is not very rigorous.

9. In Figure 4 (c), can the authors provide an estimated value of H_{DC} across the vial?

10. Will using a gradient DC field and a constant DC field lead to different results/assay sensitivity? Some literature reported a constant DC field offset on top of the AC field. The authors should comment on the differences between this work and the constant DC + AC field work.

11. The detection of antibodies is from a PBS buffer, so it's not a real clinical sample. At the end of this paper, the authors should at least comment on the clinically important concentration range of antibodies in human blood in order to prove that this sensitivity reported in this work is useful for detecting SARS-CoV-2 antibodies.

REVIEWER COMMENTS

We first want to thank all the reviewers for giving helpful comments to improve our manuscript. In the following a point-by-point answer is given with cross-references to the manuscript.

Reviewer #1 (Remarks to the Author):

This manuscript proposes a novel method called COMPASS, where a critical offset DC field is applied together with an AC excitation field in a low-cost Magnetic Particle Spectroscopy (MPS) device to elicit highly sensitive measurements of subtle changes in the hydrodynamic diameter of Magnetic Nanoparticle (MNP) ensembles. The authors also propose adding strong magnetic fields to further increase the sensitivity to detect 33 pM of SARS-COV-2-S1 antibodies, already demonstrating higher sensitivity than gold standard methods ELISA and flow cytometry, with the added benefit of much shorter protocol times (under a minute). While MPS was previously utilized to detect changes in mobility, those methods mostly relied on detecting the changes in the overall signal level. The novelty of the proposed COMPASS method lies in the fact that it does not rely on global signal changes that may be sensitive to factors such as concentration differences with respect to a reference, but rather takes advantage of abrupt phase changes in the signal at predetermined higher harmonics.

The COMPASS method is highly novel and extremely practical at the same time. It will surely have a high impact in point-of-care diagnostics and lead the way to numerous new applications. The manuscript is well written in general. My main comments are listed below, and the other points are marked in the attached PDF files (main manuscript and supplementary).

Dear Emine, many thanks for your comments and discussion.

Please find our point-by-point answers to your comments as well as your comments directly in the manuscript and supplementary part.

Main Comments:

1 - Please comment on how/whether the optimal operating parameters of the COMPASS device depend on MNP parameters and the binding vs. non-binding hydrodynamic diameter.

Dependence on particle size and magnetic susceptibility. In the revised version of the manuscript, the differences in the properties between originally (with antigen) functionalized particles' hydrodynamic diameter versus particles that have bound the complementary antibody are compared. The particles themselves are multicore particles whose crystallites are about 12 nm in size (see TEM image). Multicore particles behave differently than single-core particles of the same size, which is why the size increase obtained by binding the antibodies is to be evaluated differently than in the case of single-core particles, namely that it can be detected more sensitively. In addition, the rigid layer structure of the APTES particles is decisive for the magnetic differences that can be sensitively registered.

However, so far, we think that optimal parameters are small as possible but with a blocked magnetization. The second point is the density of linker molecules on the surface depending on the linker type (antigen, antibody, etc.). Thus, we expect a tradeoff between particle size and linker type (linker density) on the amplitude of change in effective viscosity. The ratio between particle size plus linker molecule size (hydrodynamic diameter) and binding compartment has to be optimized, e.g., by using nanobodies.

Furthermore, agglomerating particles via multiple bindings will have a higher impact on the signal change but hampers the flexibility of possible applications. We added additional information about these points in the supplementary S2.

TEM image of typical APTES-MNP

2 - The iron concentration in 25 uL of MNP-APSTES-S1 dispersion was reported as 100 ug Fe/mL, corresponding to about 2.5 ug of Fe. This is a relatively decent amount of iron, which must help provide a decent SNR for COMPASS measurements, further increasing its robustness. Please comment on whether this is a realistic amount of iron, or whether real-life point-of-care diagnostics would require the samples to be prepared at smaller volumes and/or smaller concentrations.

Indeed, this is a feasible concentration. Alternate concentrations did not result in better SNR. Higher concentrations catch lower amounts of targets per particle while lower concentrations yield less signal to work with. There should be a tradeoff between particle size, MNP concentration, and conc. of the binding compartment, which is work in progress to find out. We added additional information about these points in the supplementary S2.

3 - Supp. S2: “minimal difference of about 5% in their ζ -parameter”. Is it known how different the hydrodynamic diameter is for the binding sample vs. the reference sample in the experiments? Is it more than 5%?

The given hydrodynamic diameter of 330 nm is averaged diameter and can differ due to variations in the entire binding situation and other parameters like agility and relaxivity count. The estimated change in hydrodynamic particle diameter is the size of an antibody, which has a size of about 10 nm. This results in a naïve volume change ($=\zeta$ parameter) of about 19% ($350^3/330^3$). The actual change of ζ is likely smaller since the magnetization cannot be assumed to be strictly blocked inside with respect to cluster particles. We added more information in the supplementary part S2.

4 - Results in Fig. S2-1 and Fig. S2-2: These simulation results are dependent on the MNP parameters and H_AC parameters. For repeatability purposes, important parameters of this simulation should be provided.

The graphs in Fig. S2-1 & Fig. S2-2 show the simulation results from a quantitative simulation study based on the given Langevin-equation (EqS2-1) within a home-built simulation framework [supplementary reference 13].

In the following excerpt of the source code, the calculation process is shown: the software can calculate multiple MNPs (magnetic nanoparticle array), which are constructs holding all information of an ensemble of particles, e.g., type of magnetization (Langevin-function, Langevin-Equ, etc.) or hydrodynamic diameter, and many more.

For each MNP, specific positions in space can be defined to generate specific particle distributions, e.g., for complex structures. Based on a given coil setup, the magnetic field \mathbf{B} for each MNP position is determined and used for further calculation. In the given snippet, the magnetization of one single MNP entry is shown for calculating the magnetization following the Langevin-equation. With a

given number of particles = $\text{length}(_lastM)$, the given inverse zeta-parameter = $_zetainv$, a variance in $_zetaparameter$ for each particle = $_zeta_var[]$, and the stochastic $_brown$ parameter, the summation of all particles is calculated to get the change of magnetization dm per time step.

To get the entire data, for each $_zetainv$ parameter {} and each offset field step, here 200, one data set is generated and the desired harmonic information (here for the third harmonic) is extracted, processed and visualized.

For the simulations, the following parameters have been used:

number of particles: $_lastM = 5000$
 $_zetainv = m/\zeta$ [rad/mT/s] = {4; 4.8}
 $_zeta_var[] = 20\%$ variance
 $_brown = 0.0001$ [rad/s]
 $_samplingrate = 1000000$ [samples/s]
 $_datalength = 2000$ samples
AC-field = 40 mT @ 20kHz
DC-field = 0..40 mT in 200 steps

Excerpt from source code

Source: MFS5[V5.22.79:built5 (19.08.2022)]
 UConductorSim.pas: line 5399 ff.)

Inputs:

$_brown$: Double -> value for scaling Brownian rotational diffusion
 $_zetainv$: Double -> inverted zeta-parameter
 $_zeta_var[]$: array of Integer of array of Double -> consists for each particle a random zeta variation
 $_lastM[]$: array of Integer of array of vector3D -> consists of all prior magnetization vectors for each ensemble (MNP)
 $_B$: vector3D -> magnetic field at the position of the calc ensemble (MNP)

procedure Calc_IndValue_NEW (...)

begin

...

// Langevin-Equ: $m(n+1)=m(n)+zeta^{-1}*((mx_B)xm+Wxm)=(((mx_B)+W)xm)*zeta^{-1}$

8: **begin**

// get number of all particles used for calculating the magnetization of

// this ensemble (MNP)

$_lM := \text{length}(_lastM)$;

$dm := \text{System.Math.VectorsDouble.Vector3D}(0, 0, 0, 0)$; // init

$_brown := _brown / \text{sqrt}(\text{global_samplingrate})$;

case global_LANGEVIN_MT_SPLIT **of**

1: **begin** // single threaded

for $i := 0$ **to** $_lM - 1$ **do begin**

$\omega := (_zetainv + _zetainv*_zeta_var[i])$
 $\quad * _lastM[i].\text{CrossProduct}(_B)$;

$brown := _brown * \text{get_rnd_vector}()$;

```

    dm := brown + omega.CrossProduct(_lastM[i]);
    dm.W := 0;

    _lastM[i] := (_lastM[i] + dm).Normalize;
    dm := dm + _lastM[i];
  end;
end;
else begin
  ... // multi threading implementation
end;
end; // case

end;

...
end; // end of procedure

```

The parameters for the particle system are inspired from the APTES-MNP particle system and are given to

$$m=4 \cdot 10^{-16} \text{ Am}^2$$

$$\eta=1 \text{ mPa}\cdot\text{s}$$

$$\zeta=\kappa \cdot \eta \cdot R^3$$

$$\kappa=3$$

$$R=330 \text{ nm}$$

In the simulation, the magnetization is normalized ($m=1$) and the `_zetainv` parameter used in the simulation has to be normalized by the sampling rate (SR).

Thus, the important information for a successful simulation are:

$$m/\zeta=4 \cdot 10^6 \text{ 1/s, which can be seen as 'quality factor'}$$

$$\text{brown}=\sqrt{(2k_B T/\zeta/\Delta t)}=1 \cdot 10^{-4}$$

We added more information in the supplementary part S2.

5 - The simulations utilized offset DC fields in the same axis as the AC field, whereas Fig. S5-2 shows that the offset DC field from the permanent magnets is at an angle to the AC field. The authors then test two different configurations for the magnet in Fig. 5b, but neither of these cases coincide with the simulated case. Please comment on whether it is better for the DC field to be parallel or orthogonal to the AC field.

That is an important question. From an ongoing simulation study, we can say at the moment, that there is a slight difference in the signals for different offset field orientation, but it does not seem essential for the understanding the COMPASS effect. The SPION sample 'sees' a higher offset field, but the projected offset field on the signal acquisition as well as the AC field axis is mainly responsible for the effect (with some corrections).

For the initial simulations (S2-1 & S2-2), the offset magnetic field is parallel to the AC field, following the measurements of full data sets given in S4.

The important point in Fig. 5 was to demonstrate our findings at the beginning, that with different offset fields, the signal varies dramatically. Another important fact is, that with a gradient DC field, the robustness can be increased as the simulation in S2-2 shows.

As of now, the understanding of the effect itself is incomplete. Thus, there are multiple simulation studies as well as measurements in progress at the moment.

6 - Supp. S4: The authors mention that combining phase and amplitude can provide further increase in the difference between the binding vs. non-binding samples. In this work, the reported sensitivity of 33 pM for COMPASS was based on the phase only. I understand that this manuscript presents the general framework of COMPASS, already providing extensive analysis on numerous parameters, which is highly appreciated. Is there a reason the amplitude effect was not incorporated in this work? Are there any disadvantages to incorporating the amplitude (e.g., potential sensitivity to concentration variations)? Please comment.

That is also a good question, and we have to refer to the previous comment, that we are trying to understand this effect in more detail. The point is, that in the vicinity of the CPs, the phase changes quite rapidly, but also the amplitude drops. In the case of a differential measurement, the diff-amplitude shows a non-zero value. We think that these diff-amplitude values could be used as a measure of reliability in the future. We do not know at the moment, if the amplitude provides independent information that could be used for additional sensitivity enhancement. This is still work in progress.

7 - Supp. S5: Calibration steps for COMPASS do not need any measurements with a binding sample. This sounds like a huge advantage and should be further emphasized. With that said, would the calibration improve if it incorporated measurements with both a reference sample and a binding sample (e.g., at high AB concentration)? Please comment.

That is a good point. We are also working on that question to figure out that point. At the moment, for measurements, the COMPASS device is set up on a new reference sample (adjusting AC/DC ratio for a dedicated CP). With that calibration procedure, the presented results have been performed.

The advantage of course for that approach is the robustness and independence of external influences (temperature, humidity, personnel, etc.).

We think about new ways, which use multiple calibration samples, e.g., by adding a defined amount of (non-)binding compartments in the serum, but as mentioned, this is still work in progress.

We also added a more realistic experiment with blood serum samples (please see reviewer 2 comment 4).

Emine Ulku Saritas

Reviewer #2 (Remarks to the Author):

Several approaches to realize highly sensitive homogeneous bioassays based on functionalized magnetic nanoparticles (MNP) as markers have been reported for the detection of proteins, antibodies etc. Generally, the change of the Brownian relaxation time caused by an increase of the hydrodynamic particle size upon binding of analyte is detected. This includes measurements of the complex (AC) susceptibility and magnetic particle spectroscopy (MPS). In this manuscript, the authors describe an MPS-based approach. In contrast to MPS-based assays by other groups, the authors apply an additional dc offset magnetic field parallel to the ac magnetic field. As a certain strength of this offset field, the amplitude of a given harmonic changes sign and the phase exhibits an abrupt change by 180° . The authors call this field “critical offset magnetic field”. Importantly and as simulations show, this critical offset magnetic field is very sensitive to changes of the hydrodynamic particle diameter, which increases upon conjugation of analyte, or medium viscosity, proposed that the MNP are thermally blocked, i.e. the Brownian mechanism dominates the dynamics. Thus, in the vicinity of the critical offset field, tiny changes of the hydrodynamic diameter cause a measurable change in phase. On the other hand, the signal amplitude at this critical offset magnetic field gets rather small, which will be even more critical if higher harmonics are considered as presented in the submitted manuscript. Although the approach is quite original, the manuscript is not suited for publication in Nature Communications. The manuscript just presents a novel approach for the realization of homogeneous bioassays based on functionalized magnetic nanoparticles but omits/ignores many details that are relevant for a practical application of the presented assay and that are required to justify the publication in such a high-ranking journal.

We want to thank the reviewer for the statements about the novelty of this effect. As noted, it is a novel approach, which shows potential but, so far, several parameters and features are not fully understood. As we stated in the manuscript, we think, that this finding can be an interesting new way for highly sensitive measurements of small changes in effective particle mobility. Since this effect is new, we have no full explanation of all features and parameters at the moment. But we can say that the effect shows a robustness, which makes it very promising for designing and deploying cheap and portable hardware. In addition, the measurement is easy-to-handle and the data can be analyzed by untrained personnel.

As the reviewer mentioned, that we “omit/ignore” many relevant details, we want to invite the reviewer for a deeper discussion to enhance the quality of our manuscript. As mentioned, multiple open questions have to be answered in the near future, maybe from a larger community.

Please find a point-by-point answer to your comments in the following.

1) The authors employ comparably large MNP (hydrodynamic diameter of 330 nm after functionalization) for the detection of rather small analytes (antibodies). Using standard expression for the Brownian relaxation time and assuming room temperature and the viscosity of water, this diameter corresponds to a characteristic frequency of the nanoparticles of about 12 Hz. For comparison, the excitation frequency amounts to 20 kHz, i.e. MNP can barely follow the excitation field. This means that the system is far from being optimized and puts the presented data into question.

Many thanks for that comment. As mentioned, the hydrodynamic diameter of the multicore particles after functionalization is about 330 nm.

Here it is important to distinguish two very different effects: first, the thermally caused particle rotation, i.e., Brownian rotational diffusion, and second, the particle rotation due to the torque the external magnetic field exerts on the magnetic moment locked in the particle, which are both independent parameters.

The critical frequency f_{crit} of a particle system is the limit where the ability to follow an external rotation of the magnetic field synchronously vanishes. This can be calculated according to equation EqnS2-1 (Langevin-equation) to

$$f_{crit} = m / \zeta \cdot B / 2 / \pi = 4 \cdot 10^6 \cdot 40 \text{ mT} / 2 / \pi = 25.5 \text{ kHz.}$$

An exemplary magnetic moment for 25 nm single crystallite magnetite particles (SHP-25, Ocean nanotech, USA) can be calculated to $2 \cdot 10^{-18} \text{ A/m}^2$ [R5]. Thus, for 330 nm the magnetic moment can be estimated up to about $2.3 \cdot 10^{-15} \text{ A/m}^2$. The actual value will be significantly smaller since those large particles are multi-core particles and it can also dependent on the applied magnetic field (used value here and in the simulation is $4 \cdot 10^{-16} \text{ A/m}^2$).

[R5] T. Kahmann and F. Ludwig, Magnetic field dependence of the effective magnetic moment of multi-core nanoparticles, Journal of Appl Phys, vol. 127(23):233901, 2020

This frequency is conservatively estimated and lies clearly higher than the characteristic frequency of the Brownian rotational diffusion.

The relaxation time of 3 ms for 200 nm (or ~13 ms for 330 nm) particles only applies for the thermally caused rotational diffusion, which drives the system back to randomly oriented magnetic moments. In contrast to that, the Brownian rotation caused by the magnetic field not only depends on the viscous friction but also the magnetic moment of the particle and the strength of the magnetic field. This rotation rate is therefore decoupled of the thermally caused rotational diffusion rate and can be vastly higher. Typical values for the magnetic moments of the particles and the magnetic field strength found in MPS applications result in torques that are much higher than the thermally caused random torques. This is necessary for driving the magnetization of magnetic particle suspensions close to saturation in order to generate higher harmonics with a high SNR. The commonly used expression “magnetic fields reduce the Brownian relaxation time” is somewhat misleading since the actual Brownian diffusion is not affected by the

presence of a magnetic field. The presence of weak magnetic field generates a particle behavior, that can phenomenologically described by a change in the relaxation constant. In case of particles with locked magnetization this change in the effective relaxation time can directly be derived from the Langevin equation.

In case of massive particles consisting of a single crystal with blocked magnetization and without surface coating, the rotation rate caused by the magnetic field is independent of the particle size since here the magnetic moment as well as the viscous friction increase both linearly with the particle volume (neglecting surface effects in magnetic materials, that reduce the magnetic moment in smaller particles compared to the bulk saturation magnetization). There are no homogeneous particles available with diameters much larger than 30 nm (at least not commercially). Larger particles consist typically of aggregates of smaller particle cores. MPS measurements of these particles indicate that they also can have a strongly locked magnetization, which is most likely caused by large anisotropies and particle interaction of densely packed particle cores, e.g, described in Ota et al “Characterization of Neel and Brownian Relaxations Isolated from Complex Dynamics Influenced by Dipole Interactions in Magnetic Nanoparticles”.

We also agree with the reviewer, that a more optimized particle system would be preferably particles with a smaller diameter. However, we are also quite surprised about the excellent results even with such large MNPs.

However, further experiments are planned for investigate this effect in more detail.

2) The authors claim that their approach is rapid with a measurement time of well below a minute (see abstract). In S8 of the SI, they even claim that the results are available with seconds. This is very questionable since in other ACS or MPS based homogeneous assays, which rely on the same basic principle, incubation times of up to hours were reported. The content of the manuscript does not consider any refined techniques to speed up the diffusion-limited binding kinetics. To experimentally demonstrate that the binding has really finished after a minute (or a few), measurements of the binding kinetics need to be added.

We did not refer to measurement times of hours, the most competitive approach in comparison to ours requires measuring times of 5 to 10 minutes [R0].

Of course, the particle system used for our experiments should also work for ACS and MPS experiments (not tested yet). The preparation for serum measurements still requires more time since the preparation of the serum cannot be accelerated. However, the measurement times is 20 ms per measurement and depending on the signal strength, there is no averaging required. Thus, the COMPASS measurement seems to be much more robust than MPS or ACS measurements.

The binding kinetics is an interesting part here and we want to show initial results measuring the real-time binding kinetics of APTES SPIONs + MERS and SARS antibodies. The graph shows the phase progression over time. The COMPASS device was calibrated on the 5th harmonic (see S5) and the phase was measured

continuously with an acquisition time of 20 ms and a delay time of 1 second between the experiments. At time point 20 and 110, 25 μ l MERS antibodies (dilution 1:50k) as well as 25 μ l SARS antibodies (dilution 1:50k) has been added. As expected, the MERS ABs only slightly change the phase, but the SARS ABs almost instantaneously change the phase dramatically and show a saturation within 75 seconds, which is comparable with the literature [R1, R2, R3]. Considering binding kinetics the 3D accessibility of nanoparticles is inherently higher than for flat surface immobilized bait proteins. As these results are far outside the scope of this manuscript, we would prefer not to include them in this manuscript.

Fig.: Real-time antibody-antigen interaction analysis using COMPASS.

[R0] Wang et al., *Ultrasensitive, high-throughput, and rapid simultaneous detection of SARS-CoV-2 antigens and IgG/IgM antibodies within 10 min through an immunoassay biochip*, *Microchimica Acta*, vol. 188:262, 2021

[R1] R. Karlsson et al., *Kinetic analysis of monoclonal antibody-antigen interactions with a new biosensor based analytical system*, *Journal of Immunological Methods*, vol. 145, pp. 229-40, 1991

[R2] A.C. Malmborg et al., *Real Time Analysis of Antibody-Antigen Reaction Kinetics*, *Scand. J. Immunol*, vol. 35, pp., 643-50, 1992

[R3] P.S. Katsamba et al – *Kinetic analysis of a high-affinity antibody/antigen interaction performed by multiple Biacore users*, *Analytical Biochemistry*, vol. 352, pp. 208-21, 2006

3) The reported limit of detection for the SARS-CoV-2 S1 antibodies is in the same range as the values reported by other groups (however, for spike or N proteins as well as for mimic viruses). Thus, there is no significant progress, which is needed to make magnetic nanoparticle-based bioassays compatible or superior to standard/existing techniques.

LOD is one argument another is the measurement time, which is by far the shortest for our approach. Moreover, the sample processing where we do not need washing steps. As mentioned above, the measuring effect is not fully understood so far, but already shows a high potential since all data have been acquired within a home-built portable and cheap system. Further optimizations of the particle system as well as the hardware design, sequencing and data processing (consideration of the amplitude around CPs) could enhance the sensitivity of COMPASS.

4) In Fig. S8 the authors suggest that the presented approach is readily usable for measurements in different media as e.g. saliva (nasal/throat swab) or blood. Experts in the field know that such a statement has to be considered with care since the chemistry in real media is very complex and produces many new problems. Thus, experimental data on real samples should be presented. Otherwise, the statement that the approach is universally usable should be omitted.

Indeed, these measurement had not been conducted for the original version of the manuscript. Therefore, our experts in this field have now implement measurements in the complex medium blood serum, which is most relevant for antibody detection. The following figures show the first results for SARS detection in blood serum. The left graph shows an initial calibration experiment demonstrating the absence of cross-interactions. For that, a pure SPION probe, SPION+serum⁻, SPION+serum⁻+SARS ABs and SPION+serum⁻+MERS ABs, have been measured 5 times each. Clearly the sample can be identified with SARS ABs.

The right graph shows a realistic measurement of three different serum samples of healthy staff members of the Institute of Virology and Immunobiology with informed consent. Serum anti-Spike IgG levels were quantified using the LIAISON® SARS-CoV-2 TrimericS IgG assay (Cut-off: 34 BAU/ml).

- *Negative serum0⁻*
- *high level serum1⁺ with 28,600 BAU/ml*
- *medium level serum2⁺ with 6,500 BAU/ml*
- *low level serum3⁺ with 44 BAU/ml*

Clearly no cross-binding effects are obtained (left figure).

In addition, all spike levels could be distinguished even to the lowest level, which is near the cut-off of 34 BAU/ml of the CLIA test.

We added these experiments in the manuscript (Fig. 7).

Furthermore, we changed Fig. S8 appropriately.

Figures show initial results of blood serum experiments with COMPASS.

5) Authors do not explicitly emphasize that the binding scheme is based on Brownian dominated nanoparticles.

We added this information in the main text of the manuscript.

6) Figure such as Fig. 6 should also be shown for the S- sample to quantify the amount of unspecific binding. This is hardly discernible from Fig. 5. Unspecific binding is a major issue in such assays and needs to be studied in more detail. In addition, it should be quoted what sample was taken for the data in Fig. 5.

That is right, unspecific binding effects can provide false-positive results. We now provide to all measurement the S- binding protocol, and added the reference measurements to the new Fig. 6, where we show our latest results with a sensitivity of 1:500k \rightarrow 7 pM. Still no cross-binding effects are visible.

In addition, a more realistic experiment has been added in the manuscript with blood serum (see Fig. 7), also showing no cross-binding effects.

Furthermore, we added additional information in the caption of Fig. 5.

7) In the abstract and later in the text, the authors quote the sensitivity with 0.85 fmole/50 μ l sample volume, which is claimed to correspond to 33 pM. Simply taking the former numbers, one gets 17 pM. This factor of two difference needs to be explained.

Indeed, this was displayed confusing. 33 pM sensitivity is derived from considering the entire measuring solution which consists of the antigen containing sample and an equally volume of functionalized MNP containing no antibodies. Just considering the volume of 25 μ l analyte volume would correspond to 17 pM. We follow the reviewer and now refer just to the analyte sample and claim 17 pM as sensitivity.

Furthermore, we could enhance with latest measurements the sensitivity to 1:500k, which now corresponds to 7 pM as indicated in Fig. 5.

8) One critical issue is that operating the system at the critical offset magnetic field increases on one hand the sensitivity of the phase to tiny changes of the hydrodynamic size, but on the other hand, the amplitudes of the harmonics get very small. Thus, the signal-to-noise may get too small. This point, which gets even more dramatic if higher harmonics are considered, is important since the detection of tiny amounts of analyte requires very small MNP concentrations. But such information is completely missing.

That is an important point. The amplitude difference also shows extrema along the offset magnetic field, which may be a limitation. But the extrema occurs at different positions than the critical points (see Fig.3 or Fig. S4-2), since slightly changes of the Chebyshev-like curve changes along offset field axis for two measurements.

This show, that the SNR does not completely drop at these measuring points. However, to our knowledge it is true, that for high sensitivity a specific MNP

concentration is required to ‘see’ the higher harmonics in the signal. As the reviewer mentioned, this hampers the detection limit of tiny amounts of analytes. In the vicinity of a critical point, the DC offset field provides a kind of background suppression, where background can be defined as signal from particles unaltered in their mobility in the presence of the analyte.

Hence, at the CP the signal background of the unbound particles is suppressed even at high MNP concentrations. Thus, the tiny amount of MNPs that change their diameter will lead to a detectable change in the phase near the CP. For DC fields away from the CP, the measured signal is dominated by the unaltered MNP background and thus the phase change of the total signal due to the binding particles is much less pronounced.

Unfortunately, this effect cannot be fully explained at the moment. However, the effect can be numerically simulated using the Langevin equation as described in S2 and shows similar quantitative results as the experiments.

But it still offers room for more investigation.

9) p. 1, right column: The fact that ACS is complicated by long acquisition times, is only partly true. ACS can also be speeded up if measurements are performed at a single, optimum frequency (see susceptibility reduction technique by the Horng group in Taiwan). Also, the claim that ACS requires experienced personal is not correct. Performing an ACS measurements is not more complicated than carrying out MPS measurements.

We want to make this point more clear: MPS and also ACS measurements are actually easy to handle, but the signal interpretation still requires experienced personnel from our point-of-view aa opposed to COMPASS.

We reworked this part in the manuscript and removed our statement.

The reviewer is correct in stating, that ACS can be accelerated (following the IMR method by the Horng group), but this will sacrifice sensitivity.

However, as seen in the figure taken from “Toward Rapid and Sensitive Detection of SARS-CoV-2 with Functionalized Magnetic Nanoparticles” by Zhong et al., the signal in MPS measurements is given as a ratio of the 3rd to 1st (Figure 3) in dependency of the frequency (higher harmonics). In the amplitude, slight changes can be seen, which are interpreted as signal changes coming from changes of the serum concentration. However, the absolute harmonic signal in MPS experiments strongly depend on MNP concentration and requires a well monitored sample preparation to avoid a signal change coming from concentration variations. In addition, taking the first harmonic of an MPS experiments requires a well calibrated and stable MPS hardware since the direct feedthrough of the excitation signal into the receive chain have to be suppressed by filtering or gradiometric hardware.

In ACS experiments, the signal (here the relaxation time τ_B) is taken from fitting the data with an appropriate model. This processing step can be automatized but should be monitored.

However, both modalities provide good results, but we think the data processing cannot be monitored or correctly interpreted by untrained personnel. The signal interpretation with a COMPASS device is easier through the differential measurement of sample Vs reference.

[redacted]

***Figure: left:** MPS results (figure 3) taken from “Toward Rapid and Sensitive Detection of SARS-CoV-2 with Functionalized Magnetic Nanoparticles” by Zhong et al. **Right: ACS results** (figure 2) taken from “Toward Rapid and Sensitive Detection of SARS-CoV-2 with Functionalized Magnetic Nanoparticles” by Zhong et al.*

10) S1 and S2 in SI: Here the authors describe the modelling of the proposed assay.

First, it would be important to add particle and measurement parameters that they applied in the simulation. The only explicit information they provide is that the hydrodynamics diameter is varied by 1.7%.

Second, when discussing Brownian and Neel relaxation in S2, the authors quote a threshold diameter for the transition from Brownian to Neel relaxation of 30 nm. This critical diameter depends on various parameters, such as medium viscosity, shell thickness, temperature, effective anisotropy constant, and magnetic field amplitude.

That is an important point here.

*To answer the **first part**, we want to refer here to the answers of reviewer 1 comments 1 and 4.*

We additionally adapted the simulation values, as described, to be closer to the measurement and added more information in the supplementary part S2 to make this point more clear.

*For the **second part**, yes, the actual cross-over diameter where the Néel relaxation time exceeds the Brownian relaxation time depends on many parameters. The 30 nm were meant as a rough estimate above which Néel relaxation can be typically neglected for magnetite particles according to the literature. We changed this part in the supplementary part S2 appropriately.*

Reviewer #3 (Remarks to the Author):

In this manuscript, the authors reported the COMPASS method for the detection of SARS-CoV-2 S1 antibodies. This COMPASS method circumvented the limits of traditional ACS and MPS methods that also exploit the magnetization responses of MNPs to minimal changes in the mobility of MNP assemblies. The authors have explained the mechanism of 1DC+1AC magnetic field-based COMPASS method in detail. There are some concerns that the authors need to address before further consideration for publication.

1. Critical point by its nature is extremely sensitive to the applied DC field. In that case how precision in permanent magnet placement is being taken care of? Any small deviation will result in drastic change to the results as pointed in Fig.2.

Yes, that is true. Slight changes yield a dramatic change in the signal. However, consequently the hardware must be mechanically stable, also temperature issues regarding the coil system should be avoided (achieved by short pulse lengths of 20 ms per experiment every second -> and a small duty cycle of 5%). But the biggest improvement in robustness is achieved by applying a DC gradient over the measurement chamber. In addition, for each experiment, a new reference signal is acquired to be sure to get rid of signal drifts.

2. On the same note, information should also be pointed if the DC offset required for creation of higher harmonics differ from one another (i.e. 5th, 7th, 9th from 3rd), and if so, what harmonic has the device been optimized for concerning bioassay applications?

Yes, for each higher harmonics' critical points, the DC field differs as pointed out in the supplementary S5.

3. In different figures, the phase modifications of different higher harmonics are being used (refer to Fig. 5, 6, and S7-2). Considering from the end-application perspective, what is the decision-making strategy:

a. Considering phase change of one particular harmonic? If so than what is the harmonic of interest?

Good point. As we mentioned in S5, before starting an experiment, the system has to be calibrated (choose AC/DC ratio) to a desired CP. At the moment, this has been performed manually, but a fully electrical calibration is work-in-progress. However, the different CPs and harmonics we investigated show hardly any relevant performance differences.

b. Considering a phase change in all captured higher harmonics (till 11th) and looking for a phase modification in any and all of them?

Interesting point. By applying a strong DC gradient along the measurement chamber, multiple CPs can be covered resulting in signal changes on multiple CPs as indicated in Fig. S5-4.

We also think about that point to enhance the sensitivity of our measurements, but it is still work-in-progress.

4. For the results depicted in Fig. 5, What dilution/ concentration of antibodies does the test correspond to? Also, the corresponding MNP concentration should also be noted.

Thanks for the comment. The dilution is 1:10k.

We added more information about this in the manuscript.

Please also see the answers to the comment 6 of reviewer 2.

5. For the reported magnetic field in the range of 17mT, with the coil parameters given, the device would have been utilizing a current of roughly 8A. Is heating of coils and hence the MNP sample present within a concern with regard to the setup? If so, what necessary steps are included to negate this impact?

Heating is still an issue in systems working with air-coils for generating high magnetic fields. We avoid this problem by operating the portable COMPASS device with a low duty-cycle of 20 ms/1000 ms=5% (pulsed magnetic field generation).

This ensures a stable operation also with high number of measurements.

Please also see the answer to comment 1.

We added more information about this point in the supplementary part S5.

6. There is one error in the annotation in this sentence: “At a specific magnetic field strength Msat, all particles are aligned, and the magnetization of the sample is saturated (saturation magnetization Msat)”. The specific magnetic field strength is Hsat not Msat.

Corrected.

7. The experiment condition is not described in detail. For example, “After adding the antibody dilution or buffer (reference sample), the samples were mixed shortly by pipetting and directly measured without any further incubation time”. It’s not clear how much time is waited during the pipetting step before the measurement. The authors should at least provide the estimated time.

The preparation time for the samples is in the range below 1 minute depending on the working speed. As we mentioned in the manuscript, no further preparation steps are required (washing or incubation). Thus, the sample is pipetted in the APTES-MNP S1 dilution and ready for measurement within seconds.

Please also find the answer to reviewer 2 comment 2 saying, that the interaction time between antigen and antibody is in the range of few seconds. We added more information in the manuscript.

8. A followup question from my last comment. Typically, for antibody-antigen interaction, it takes dozens of minutes up to an hour for the bindings to reach equilibrium. If the mixing time is too short, I doubt the bindings are not complete yet. Even a small mixing time difference between samples to samples will result in different binding stages. Imagine the “S” shape binding curve, if the mixing time is too short and at the steep slope of the “S” shape curve, then even a small difference in the mixing time can cause a big difference in the binding results. So, I feel the experiment design here is not very rigorous.

Please see the answer to our previous comment #7 and also the answer to comment 2 of reviewer 2.

9. In Figure 4 (c), can the authors provide an estimated value of H_{DC} across the vial?

As indicated in Fig. S5-2 supplementary, it is (8±2) mT. We reworked Fig. S5-2.

10. Will using a gradient DC field and a constant DC field lead to different results/assay sensitivity? Some literature reported a constant DC field offset on top of the AC field. The authors should comment on the differences between this work and the constant DC + AC field work.

Unfortunately, we cannot see, which literature the reviewer refers to. If the reviewer is referring to ACS literature we cannot give an appropriate comment on that question since there are huge differences in AC and DC field strengths between ACS and COMPASS. Thus, it is quite difficult to be compared. As mentioned above, a strong DC gradient yields a more robust signal. However, without a DC gradient, the sensitivity should increase but also is much more sensitive to noise. So far, we cannot give a full explanation about all features of the critical point effect.

11. The detection of antibodies is from a PBS buffer, so it's not a real clinical sample. At the end of this paper, the authors should at least comment on the clinically important concentration range of antibodies in human blood in order to prove that this sensitivity reported in this work is useful for detecting SARS-CoV-2 antibodies.

That is true. Thus, we performed additional experiments with different blood serum from Covid patients and add the results in the supplementary S8.

REVIEWERS' COMMENTS

Reviewer #1 (Remarks to the Author):

The authors have thoroughly addressed all of my comments from the first cycle. In addition, the new results in blood serum (as requested by Reviewer 2) further provided a proof of the real-life practicality of COMPASS. I recommend the acceptance of this manuscript.

Emine Ulku Saritas

Reviewer #2 (Remarks to the Author):

The authors performed a thorough revision of their manuscript according to the points raised by the reviewers. Most points were satisfactorily considered. In some cases, it would have been good to provide this information/feedback not just in the response letter but also in the manuscript and SI material, respectively.

In the manuscript, the authors mainly added the signal of the non-binding sample in Fig. 6 in order to illustrate the absence of non-specific binding. Furthermore, they added Fig. 7 which shows experiments with blood serum to demonstrate the performance and robustness in a more realistic environment when focussing on the antibody detection. Both clearly strengthen the impact of their work.

Most points of criticism were accounted for in the SI document. This mainly regards more quantitative information about the suitability of the used magnetic nanoparticle system.

Nevertheless, before getting accepted for publication in Nature Communications, the manuscript should be further revised by the following points:

1. p. 6, bottom of left column: Here the authors write that the protocol includes just seconds including incubation time as indicated in Fig. 6. The incubation time does not play any role in Fig. 6. There the authors just show the signal of subsequently measured reference, S+ and S- samples. Obviously, samples were prepared prior to the measurements so that the exact time between adding analyte to the nanoparticle suspension is not addressed. Incubation time was one major point of criticism by the reviewer in his original report. The authors added a figure "Real-time antibody-antigen interaction analysis using COMPASS" to the response letter. From this figure, it becomes quite clear that binding finishes after about 75 sec. But this contradicts with the statement in the manuscript that the measurements just last a few seconds including incubation time. Since the authors performed these preliminary experiments (shown in the response letter) and know better, this statement should be modified since it promises unrealistic facts. But even incubation times of around one minute would be great. In addition, the statement in the text does not match that given in the caption of Fig. 6 where it says that "the prior preparation time including mixing and incubation of samples and particles was below 1 minute". Since there are preliminary data on the incubation kinetics, the reviewer suggests to present them as preliminary data in the SI file. This would avoid extensive discussions/speculations.
2. p. 2 of the SI document, left column: Here the authors try to quantify their simulation results, which was also a point of criticism (authors presented in the original manuscript and in the supporting material simulation results without presenting the simulation parameters). Table S2 provides the data used in the simulations. First, the magnetic moment of the applied multicore nanoparticles seems to be much too high. Clearly, the relaxation time in the high-field limit is much shorter than in the diffusion limited small-field range so that a frequency of 12 Hz is unrealistic. In the response letter the authors refer to a paper by Kahmann and Ludwig but address only the data obtained for single-core SHP25 nanoparticles. This paper also discusses the effective magnetic moment of multicore nanoparticles. For example, for BNF80 a magnetic moment in the upper 10-18 Am² range was reported. Also, for single-core Ni nanorods with core length of around 240 nm, a moment in the mean 10⁻¹⁷ Am² range was found. An effective magnetic moment of the used APTES-MNP-SBA-S1 nanoparticles in the mean 10⁻¹⁷ Am² range seems more realistic (even for hydrodynamic diameters of 330 nm). Ideally, it should be determined by fitting measured magnetization-vs.-

magnetic field curves with the Langevin function. Applying this value, one arrives at a characteristic/critical frequency in the range of a few kHz, which is somewhat too small for the excitation frequency of 20 kHz. Second, in the SI information this discussion is performed on a rather qualitative level. Why not explicitly showing the equation for the characteristic (critical) frequency provided in the response letter? This would be much more convincing.

3. Table S2: In the left column, the quantities are defined. What does "Brownian diffusion" mean? Apparently, it is an inverse time (or frequency). In the right column, the authors name the quantity "brown" which also sounds strange. Please, revise!

4. p. 4, right column: There it says that "the graph shows the phase... on selected harmonics (n=2nd to 9th)". It should mean "3rd to 9th".

5. The manuscript contains a few more typos. The authors should check the manuscript and SI document carefully again.

REVIEWERS' COMMENTS

Reviewer #1 (Remarks to the Author):

The authors have thoroughly addressed all of my comments from the first cycle. In addition, the new results in blood serum (as requested by Reviewer 2) further provided a proof of the real-life practicality of COMPASS. I recommend the acceptance of this manuscript.

Emine Ulku Saritas

Reviewer #2 (Remarks to the Author):

The authors performed a thorough revision of their manuscript according to the points raised by the reviewers. Most points were satisfactorily considered. In some cases, it would have been good to provide this information/feedback not just in the response letter but also in the manuscript and SI material, respectively.

In the manuscript, the authors mainly added the signal of the non-binding sample in Fig. 6 in order to illustrate the absence of non-specific binding. Furthermore, they added Fig. 7 which shows experiments with blood serum to demonstrate the performance and robustness in a more realistic environment when focussing on the antibody detection. Both clearly strengthen the impact of their work.

Most points of criticism were accounted for in the SI document. This mainly regards more quantitative information about the suitability of the used magnetic nanoparticle system.

Nevertheless, before getting accepted for publication in Nature Communications, the manuscript should be further revised by the following points:

1. p. 6, bottom of left column: Here the authors write that the protocol includes just seconds including incubation time as indicated in Fig. 6. The incubation time does not play any role in Fig. 6. There the authors just show the signal of subsequently measured reference, S+ and S- samples. Obviously, samples were prepared prior to the measurements so that the exact time between adding analyte to the nanoparticle suspension is not addressed. Incubation time was one major point of criticism by the reviewer in his original report. The authors added a figure "Real-time antibody-antigen interaction analysis using COMPASS" to the response letter. From this figure, it becomes quite clear that binding finishes after about 75 sec. But this contradicts with the statement in the manuscript that the measurements just last a few seconds including incubation time. Since the authors performed these preliminary experiments (shown in the response letter) and know better, this statement should be modified since it promises unrealistic facts. But even incubation times of around one minute would be great. In addition, the statement in the text does not match that given in the caption of Fig. 6 where it says that "the prior preparation time including mixing and incubation of samples and particles was below 1 minute". Since there are preliminary data on the incubation kinetics, the reviewer suggests to present them as preliminary data in the SI file. This would avoid extensive discussions/speculations.

We agree with the reviewer and added the real-time antibody-antigen interaction analysis results in the SI of the manuscript (see supplementary S8) to avoid extensive discussions.

The incubation time finishes after 75 seconds; however, the method only requires the signal change to reach a certain threshold. In this case the preliminary data shows a signal change of more than 10 standard deviations within seconds for an antibody concentration of 1:50k (20 ng/ml).

In addition, we changed the manuscript appropriately.

2. p. 2 of the SI document, left column: Here the authors try to quantify their simulation results, which was also a point of criticism (authors presented in the original manuscript and in the supporting material simulation results without presenting the simulation parameters). Table S2 provides the data used in the simulations. First, the magnetic moment of the applied multicore nanoparticles seems to be much too high. Clearly, the relaxation time in the high-field limit is much shorter than in the diffusion limited small-field range so that a frequency of 12 Hz is unrealistic. In the response letter the authors refer to a paper by Kahmann and Ludwig but address only the data obtained for single-core SHP25 nanoparticles. This paper also discusses the effective magnetic moment of multicore nanoparticles. For example, for BNF80 a magnetic moment in the upper 10-18 Am² range was reported. Also, for single-core Ni nanorods with core length of around 240 nm, a moment in the mean 10⁻¹⁷ Am² range was found. An effective magnetic moment of the used APTES-MNP-SBA-S1 nanoparticles in the mean 10⁻¹⁷ Am² range seems more realistic (even for hydrodynamic diameters of 330 nm). Ideally, it should be determined by fitting measured magnetization-vs.-magnetic field curves with the Langevin function. Applying this value, one arrives at a characteristic/critical frequency in the range of a few kHz, which is somewhat too small for the excitation frequency of 20 kHz. Second, in the SI information this discussion is performed on a rather qualitative level. Why not explicitly showing the equation for the characteristic (critical) frequency provided in the response letter? This would be much more convincing.

We follow the reviewers' suggestion to include a short paragraph for the critical frequency f_{crit} in the SI of the manuscript (supplementary S2).

We agree with the reviewer that the value of the magnetic moment for BNF80 particles suggests, that the actual magnetic moment of the used particles is probably lower than the assumed value in our exemplary simulation. The actual particle signal suggests a higher magnetic moment based on the magnitude of the generated higher harmonics, which is most likely due to "slippage", i.e.; the magnetization sees a reduced viscous friction since it can also rotate inside of the particle (this sounds like Néel relaxation, but in multi core particles this is a more complicated multi particle phenomenon). The result is higher magnitudes for higher harmonics as compared to particles with the same size and magnetic moment but with fully fixated magnetization. At the same time the stronger signal will be less affected by changes in the hydrodynamic diameter of the particles. It is therefore very likely, that the method can be significantly improved by using more optimized particles.

3. Table S2: In the left column, the quantities are defined. What does "Brownian diffusion" mean? Apparently, it is an inverse time (or frequency). In the right column, the authors name the quantity "brown" which also sounds strange. Please, revise!

That is true. "Brownian diffusion" is misleading here and has been corrected into "Magnitude of stochastic term in EqS2-1".

4. p. 4, right column: There it says that "the graph shows the phase... on selected harmonics (n=2nd to 9th)". It should mean "3rd to 9th".

Corrected.

5. The manuscript contains a few more typos. The authors should check the manuscript and SI document carefully again.

The manuscript has been reworked.